# High accuracy UAV photogrammetry of ice sheet dynamics with no ground control

Thomas R. Chudley[1], Poul Christoffersen[1], Samuel H. Doyle[2], Antonio Abellan[3], and Neal Snooke[4]

[1]Scott Polar Research Institute, University of Cambridge, Cambridge, UK
[2]Centre for Glaciology, Department of Geography and Earth Sciences, Aberystwyth University, Aberystwyth, UK
[3]Institute of Applied Geoscience, School of Earth and Environment, University of Leeds, Leeds, UK
[4]Department of Computer Science, Aberystwyth University, Aberystwyth, UK

**Correspondence:** Thomas R. Chudley (trc33@cam.ac.uk)

**Abstract.** Unmanned Aerial Vehicles (UAVs) and Structure from Motion with Multi-View Stereo (SfM-MVS) photogrammetry are increasingly common tools for geoscience applications, but final product accuracy can be significantly diminished in the absence of a dense and well-distributed network of ground control points (GCPs). This is problematic in inaccessible or hazardous field environments, including highly crevassed glaciers, where implementing suitable GCP networks would be logistically difficult if not impossible. To overcome this challenge, we present an alternative geolocation approach known as GNSS-supported aerial triangulation (GNSS-AT). Here, an on-board carrier-phase GNSS receiver is used to determine the location of photo acquisitions using kinematic differential carrier-phase positioning. The camera positions can be used as the geospatial input to the photogrammetry process. We describe the implementation of this method in a low-cost, custom-built UAV, and apply the method in a glaciological setting at Store Glacier in West Greenland. We validate the technique at the calving front, achieving topographic uncertainties of $\pm0.12$ m horizontally ($\sim$1.1x the ground sampling distance) and $\pm0.14$ m vertically ($\sim$1.3x the ground sampling distance) when flying at an altitude of $\sim$450 m a.s.l. This compares favourably with previous GCP-derived uncertainties in glacial environments, and allowed us to apply the SfM-MVS photogrammetry at an inland study site where ice flows at 2 m d$^{-1}$ and where stable ground control is not available. Here, we were able to produce, without the use of GCPs, the first UAV-derived velocity fields of an ice sheet interior. Given the growing use of UAVs and SfM-MVS in glaciology and the geosciences, GNSS-AT will be of interest to those wishing to use UAV photogrammetry to obtain high-precision measurements of topographic change in contexts where GCP collection is logistically constrained.

## 1 Introduction

In recent years, Unmanned Aerial Vehicles (UAVs) have emerged as a versatile and practical tool for aerial surveying. A common application of this method that holds particular promise in the geosciences is the production of 3D topographic models from sequential 2D imagery using Structure from Motion with Multi-View Stereo (SfM-MVS) photogrammetry (Westoby et al., 2012; Fonstad et al., 2013; Eltner et al., 2016). With repeat surveys enabled through flight autonomy, SfM-MVS is creating new opportunities for the study of terrain evolution in 4D (James et al., 2017). The technique compliments, and provides key advantages over, satellite-based earth observation methods, which have larger spatial coverage but lower spatial

resolution, as well as an inherent trade off between spatial and temporal resolution in many applications. With a relatively low barrier of entry in terms of cost, UAV-derived photogrammetry is rapidly advancing and the versatility of the technique provides new avenues of research using additional image processing methods or on-board sensors, many of which have yet to be explored. UAV-SfM has become an increasingly used tool within the cryospheric sciences (see Bhardwaj et al., 2016), in particular through the application of feature-tracking methods to multitemporal datasets in order to produce velocity datasets in glacial environments as diverse as the Himalaya (Immerzeel et al., 2014; Kraaijenbrink et al., 2016), Alps (Seier et al., 2017), Peruvian Andes (Wigmore and Mark, 2017), and the Greenland Ice Sheet (Ryan et al., 2015; Jouvet et al., 2017, 2018).

While UAV-derived photogrammetry offers key advantages over conventional surveying techniques in studies of 4D topographic change, the dependency on ground control points (GCPs) is often impractical and a hindering factor needed to scale and orient photogrammetric models to a real coordinate system (James and Robson, 2014; Carrivick et al., 2016). Previous work has shown that the quantity and distribution of GCPs can have a significant impact on the final accuracy of the photogrammetric products: for example, topographic error has been shown to increase if the number of GCPs is decreased and spacing between GCPs increases (Tahar, 2013; Johnson et al., 2014; James and Robson, 2014; Shahbazi et al., 2015; Tonkin and Midgley, 2016). Accuracy assessments performed specifically for a glaciological environment report that for a ground sampling distance (GSD) of ∼6 cm, local accuracy decreases with the distance to the closest GCP at a rate of about 0.09 m per 100 m (Gindraux et al., 2017). Additionally, Gindraux et al. (2017) report an optimal GCP distribution density (i.e. beyond which no improvement in accuracy is observed for their GSD) of 7 GCP km$^{-2}$ for horizontal accuracy and 17 GCP km$^{-2}$ for vertical accuracy. Producing a GCP network of this density in glacial terrain can be impractical, logistically-expensive to collect, and often unfeasible – as well as limiting one of the inherent advantages of UAVs in being able to remotely and accurately observe terrain which is difficult and hazardous to access on the ground. The difficulties of producing these networks can be observed in applied glaciological studies, where GCPs are often located only along the valley sides near a glacier's lateral margin (e.g. Immerzeel et al., 2014; Ryan et al., 2015). On-ice GCPs, if used, require repeat surveying as GCPs continuously advect with the glacier's flow. On fast-flowing glaciers (surface velocities of metres per day), these changes are so rapid that GCP collection would need to be nearly contemporaneous with image acquisition to be effective for accurate geolocation – a requirement which is unfeasible for these glaciers due to crevasses forming on their surface. As a result of the difficulties in building GCP networks in glacial environments, alternative methods are often applied to externally constrain photogrammetric products. Such methods include using tie points to tie datasets together geodetically (Kraaijenbrink et al., 2016); linearly interpolating the on-ice GCP location from the beginning and end of a UAV campaign (Jouvet et al., 2017); or providing some additional external constraint using an on-board navigational GPS geolocation (Ryan et al., 2015; Jouvet et al., 2017). The practical limitations of GCP collection is one of the most limiting factors in UAV-derived photogrammetry in the geosciences, especially in glaciological studies, where errors to date have been systematically larger than what is theoretically possible with this technique. Furthermore, these limitations have meant that no one has, to date, succeeded in using UAV-based methods to derive 4D surface evolution and velocity fields away from an ice sheet margin, where topographic ground-control is especially scarce and often lacking altogether.

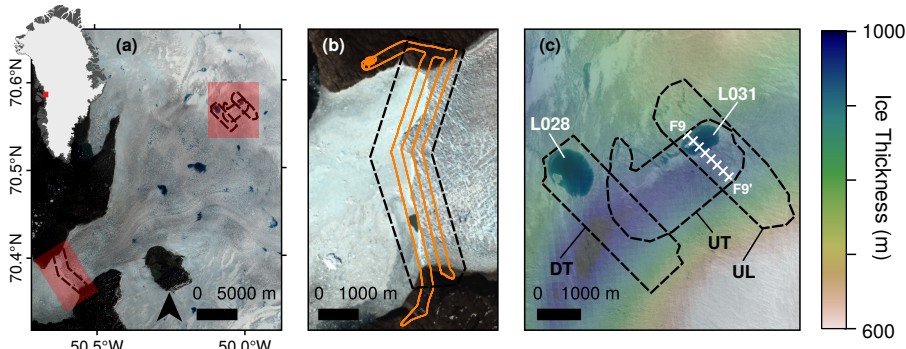

**Figure 1.** Location of study sites. (a) Store Glacier with calving front and inland study sites highlighted. Inset: location of Store Glacier in Greenland. (b) Calving front flight zone with example flight path shown. (c) Inland flight zones with labelled names: downstream transverse (DT), upstream longitudinal (UL), and upstream transverse (UT). Transect F9 marks location of fig. 9. Ice thickness from BedMachine v3 (Morlighem et al., 2017) is overlaid, and supraglacial lakes at the inland study site (L028 and L031) are also labelled.

Recent developments in lightweight, low-cost GNSS technology have allowed for the proliferation of a new technique whereby differential carrier-phase GNSS positioning is used to accurately geolocate imagery and subsequent photogrammetric products. This technique, known as GNSS-supported aerial triangulation (GNSS-AT; Benassi et al., 2017), has been shown to result in sub-GSD horizontal accuracy without the use of GCPs (Mian et al., 2015; Fazeli et al., 2016; Hugenholtz et al., 2016; Benassi et al., 2017; van der Sluijs et al., 2018). Published applications of this technique in the geosciences are so far limited (van der Sluijs et al., 2018; Strick et al., 2018), and no studies yet examine the appropriateness of this technique for the study of glacial dynamics.

The aim of this paper is to: (i) apply GNSS-AT using a low-cost, custom-built airframe suitable for the study of extreme environments; (ii) develop and describe modifications to the GNSS-AT process to allow surveys to be undertaken at inland ice sheet location far from suitable GPS reference stations; and (iii) validate the method for the study of glacier dynamics. Here, we demonstrate the suitability of GNSS-AT assisted UAV photogrammetry for assessing glacier dynamics using examples from two specific settings where on-ice GCPs are not feasible. The first is the glacier's calving terminus, where deep fractures prohibit access, and bedrock exposure allows method uncertainty to be quantified; the second is the interior ice sheet where there is no exposed bedrock and therefore distributed ground control is prohibitively difficult.

## 2 Methods

### 2.1 Study site

Store Glacier (*Qarassap Sermia*, 70.4 ° N 50.6 ° W) is a marine-terminating outlet glacier in West Greenland. The third-fastest outlet glacier in Greenland, it has a 5.2 km wide calving front draining a ∼34,000 km$^2$ catchment (Rignot et al., 2008). The terminus of Store Glacier has been located in approximately the same position since at least 1948 (Weidick et al., 1995), likely

due to the presence of a prominent basal pinning point and the position of the terminus at a lateral valley constriction (Todd et al., 2018). The calving front of Store Glacier also marks the study site of a previous. application of UAVs to the study of glacial dynamics in Greenland by Ryan et al. (2015). Store's ice catchment extends 280 km from the calving front (Todd et al., 2018), and is underlain by an active subglacial hydrological system extending at least 30 km inland.

We surveyed two locations on Store Glacier: (i) at the calving front of Store, and (ii) at an on-ice site 30 km inland (Fig. 1). Our flights at the calving front were designed to test the GNSS-AT method, with exposed bedrock at the sides of the calving front providing good ground control for validation and error quantification. The location of our primary inland flight zones were motivated by a subglacial bedrock trough visible in Bedmachine v3 data (Morlighem et al., 2017), which our flights profile longitudinally and transversely (Fig. 1c).

**2.2   UAV platform and flight planning**

We used a Skywalker X8 UAV (Figs. 2a, S1), an off-the-shelf fixed-wing air frame with a 2.12 m wingspan (Ryan et al., 2015; Jouvet et al., 2017). In a setup similar to the one used by Jouvet et al. (2017), we use open hardware "PixHawk" autopilot (https://pixhawk.org/) and APM Arduplane firmware (http://ardupilot.org/plane/) for flight control along a pre-programmed flight path. The UAV is capable of ~1 hour of flight time at a ~60 km h$^{-1}$ cruising speed, although given our use case in an
extreme environment, we flew conservatively for no more than 40 minutes. The total scientific payload weighs 500 g. This is includes a nadir mounted Sony $\alpha$6000 24 MP camera with fixed 16 mm lens. To allow for direct georeferencing of each photo location, we included an on-board lightweight L1 carrier-phase GNSS receiver (an Emlid Reach, using a small Tallysman TW4721 antenna with a 100mm ground plane). The GNSS receiver was powered by the PixHawk autopilot, and recorded camera trigger events in the output RINEX data via a hot shoe trigger cable linked to the camera. The cost of a complete
flight unit (including frame, hardware, and scientific payload) was approximately ~£1500 per unit. Further necessary ground equipment, which could be shared between units, came to ~£300: this includes the radio transmitter and lithium polymer battery charger, but not the ground-based GPS (sections 2.3, 4.3).

The UAV flew autonomously along pre-defined flight routes designed on-site using Ardupilot's Mission Planner software. The 5m ArcticDEM mosaic (Porter et al., 2018) was used to assist with the flight path design, ensuring a constant relative
altitude over the glacier and avoiding collision with high relief topography at the glacier margins. For each flight, the UAV flew a route autonomously at a relative altitude of ~450 m above ground level, resulting in a ground-level footprint of ~660 x 440 m and a GSD of ~11 cm. Our camera was set to autofocus, and a fixed f-stop and ISO (between f/4–f/8 and ISO 100-400 respectively depending on lighting conditions) chosen to target a auto shutter speed of 1/1000 s. Photos were recorded in RAW format to ensure lossless storage of images, and converted into Photoscan-compatible 16-bit tiffs before processing. Flight
lines were spaced ~250 m apart and the camera was set to trigger every ~80 m, typically acquiring ~300 images in an average flight. These parameters ensured adequate overlap in the photographs for photogrammetry purposes, targeting 80% in the flight direction and 60% in the cross-flight direction. Flight paths in the ice sheet interior, where flight endurance allowed, also included a lower-altitude ~200 m along-track flightline with sharp banking turns designed to obtain imagery from multiple elevations and oblique angles. The aim of these lower-level flights was to reduce the potential vertical 'doming' effect on

reconstructed surface topography that can occur when using self-calibrating bundle adjustment with image sets consisting of solely near-parallel viewing directions (James and Robson, 2014; James et al., 2017; Nesbit and Hugenholtz, 2019).

## 2.3 GNSS-supported aerial triangulation

The block orientation process of SfM-MVS photogrammetry can be performed in two main ways (Benassi et al., 2017). The first is Indirect Sensor Orientation (InSO), where ground-based GCPs provide external constraints. The second is Direct Sensor Orientation (DSO, sometimes referred to as 'direct georeferencing'), where external orientation parameters are provided by on-board systems including GNSS and an inertial measurement unit (IMU). Full DSO combines camera orientation data (e.g. from the IMU) with accurate camera location data from a GNSS receiver (see Cucci et al., 2017). Although DSO is not a new method for aerial photogrammetry (e.g. Blankenberg, 1992), InSO based methods have prevailed in UAV-based surveying, as the inexpensive navigational GNSS and IMU equipped in standard commercial UAVs are not accurate enough to provide more than metre-scale accuracy (James et al., 2017). Recently, commercial off-the shelf UAV units with DSO capability have become available, although these remain expensive, often in excess of £20,000 for fixed-wing units at the time of writing. Here, we take advantage of the recent availability of low-cost, light-weight carrier-phase GNSS recievers to implement direct orientation for the first time in a glaciological study. The implementation described in this study is a subset of DSO referred to as GNSS-supported Aerial Triangulation (GNSS-AT), which requires GNSS data but not IMU data (Benassi et al., 2017). GNSS-AT is therefore well-suited to UAV applications where IMU data is not available or not accurate enough (e.g. where IMU data is limited to that from lower-quality navigational units). GNSS-AT does, however, require position data that is more accurate than that provided by the GNSS receivers typically used for UAV navigation which use the Standard Positioning Service (SPS). Higher positioning accuracy than is offered by the SPS can be achieved by using differential carrier phase positioning, which makes use of the ability of GNSS receivers to measure the carrier phase to one hundredth of a cycle, equivalent to about 2 mm in distance (Leick, 2004).

To obtain accurate camera positions we kinematically post-processed 5 Hz data logged by the on-board L1 carrier-phase GNSS receiver. Data were post-processed using the differential carrier phase kinematic program within Emlid's b27 fork of RTKLIB v. 2.4.3 software relative to a base station located at the launch site. Single-frequency receivers such as the Emlid Reach can be used for differential carrier-phase positioning for baselines on the order of kilometres – distances over which the differential ionospheric delay is negligible. To apply differential corrections over the longer baselines as is often necessary in glacial environments, dual-frequency (L1/L2) receivers must be used to cancel out the frequency-dependent ionospheric delay. As dual-frequency GNSS receivers suitable for integrating in to the UAV were not available at the time of the survey (see section 4.3) we use single-frequency carrier phase positioning to determine the camera position ('R' in Fig. 2c) relative to a nearby base station ('B$_1$'), and dual-frequency carrier-phase positioning to determine the absolute position of the base station ('B$_1$') relative to a bedrock-mounted reference station ('B$_2$'). This method has the limitation that the UAV must stay within 10 km of the launch site base station, which may be located on or off the ice, but allows the launch site base station, and therefore the UAV flight, to be located long distances away from the bedrock-mounted reference station. In this study, our base station (B1) was a Trimble R9s GNSS receiver (with Zephyr 3 antenna) located at the launch site, and the bedrock-mounted reference

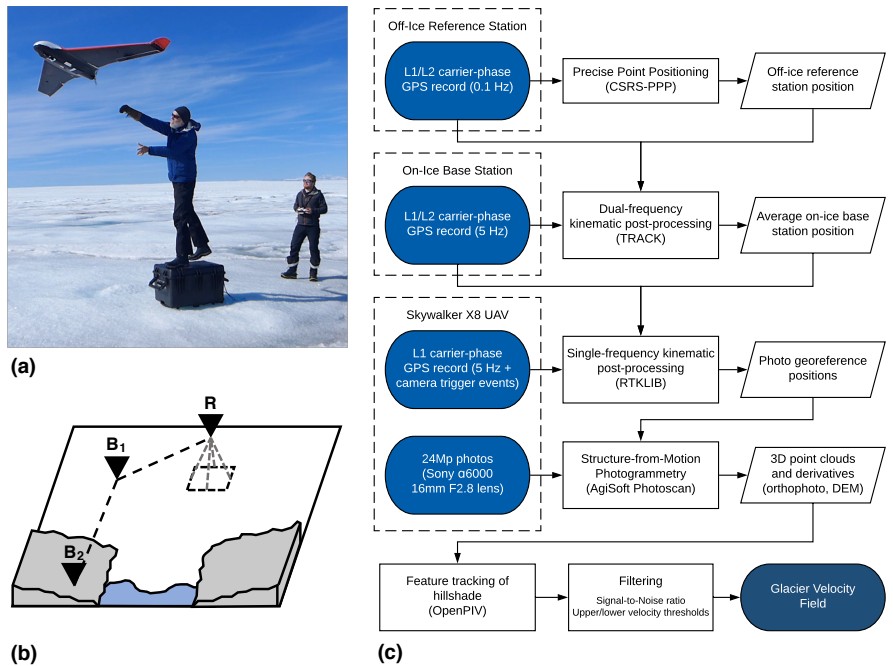

**Figure 2.** The method used in this study: (a) Launching the Skywalker X8 on the ice sheet; (b) cartoon visualising the relationship of kinematic GPS corrections between the UAV rover (R), on-ice launch site base station ($B_1$), and the off-ice reference station ($B_2$); and (c) flowchart showing the workflow used in this study to derive photogrammetry products and velocity fields at the inland study site.

station was a continuously-operating Trimble NetR9 GNSS receiver (with Zephyr 3 Geodetic antenna) recording at 0.1 Hz located at Qarassap Nunata (70.4 ° N, 50.7 ° W), a mountain ridge near Store Glacier's calving front. For practical reasons and redundancy, we used this three-receiver set-up for all flights including those at the calving front, however, only one of the dual-frequency receivers was strictly required for flights at the calving front, where a bedrock-mounted base station was located nearby.

Whilst the Emlid Reach GNSS receiver is capable of real-time kinematic (RTK) we used instead post-processed kinematic (PPK) positioning for three primary reasons. First, PPK does not rely on maintaining a reliable real-time radio link with a GNSS base station, which would introduce additional technical constraints. Second, PPK solutions are also often more accurate than RTK solutions as precise ephemeris data for the GNSS satellites is available during post processing. Third, absolute positioning using RTK requires a stationary reference station with a known position, which is not possible in real time on an advecting ice surface.

The overall workflow for photogrammetric reconstruction and for the generation of the glacier velocity field is illustrated in Figure 2c. First, the position of the Qarassap Nunata reference station was estimated using the average of 17 days of data collected at 0.1 Hz and processed with Precise Point Positioning (PPP) using the Natural Resources Canada Precise Point Positioning service (webapp.geod.nrcan.gc.ca/geod/tools-outils/ppp.php). Second, the position of the launch site base station

was determined and for this two different methods were used depending on whether the base station was located on or off the ice. Where the base station was located on bedrock its position was determined using static differential carrier-phase positioning within RTKLIB 2.4.3 software. For flights at the ice sheet interior, the launch site base station was moving at approximately 1.5 m d$^{-1}$. We therefore processed this data kinematically (King, 2004) using the differential carrier phase positioning software Track v1.30 (Chen, 1998, http://geoweb.mit.edu/gg/). All GNSS processing used final precise ephemeris products from the International GNSS Service (Dow et al., 2009). We took the average position of the base station over the flight time as the absolute reference location. During the ∼20 minute flight period the base station could have moved by up to ∼2 cm, introducing a systematic error into the final calculated photo location. Given the small magnitude of this error relative to larger errors later in the workflow, we assume the interior base station data during the flight can be treated as stationary for post-processing purposes. Finally, we post-process the UAV-based data kinematically against the launch site base station data using Emlid's RTKLIB 2.4.3 b27 fork. The Emlid RTKLIB fork provides final photo geolocation using the GPS time of the camera trigger marker in the RINEX data by linearly interpolating between the two closest points of the 5 Hz record. RTKLIB canera location outputs are estimated to have standard deviations ∼0.6 mm horizontally and ∼1.1 mm vertically for fixed solution data.

## 2.4 SfM-MVS photogrammetry and feature tracking

SfM-MVS photogrammetry was performed with AgiSoft Photoscan (version 1.3.3; http://www.agisoft.com), using the determined camera positions in the input process. As geolocation was accurate to within millimetres, it was also necessary to include the directional offset between the receiver antenna and camera position (-7.9 cm in the Y direction and +13.2 cm in the Z direction relative to the lens of the camera) to properly locate camera centre points. Camera calibration was performed automatically in the bundle adjustment process, which is the preferred option when other variables of the bundle adjustment are well constrained. From the final dense point clouds, we produce orthophotos at 0.15 m resolution and geoid-corrected DEMs at 0.2 m resolution based on recommended output resolutions from Photoscan.

We produced horizontal velocity fields by feature tracking 0.2 m resolution multidirectional hillshade models produced from the DEMs using GDAL 2.2. Using DEM-derived products has the disadvantage of having a slightly lower resolution than an orthophoto, but the advantage of being consistently comparable when tracking datasets collected in variable lighting conditions. In particular, orthophotos acquired at different times of the day can complicate feature tracking due to the variation in shadow directions (cf. Jouvet et al., 2017). To feature track images, we used OpenPIV (Taylor et al., 2010), an open-source particle image velocimetry software implemented in MATLAB. Following a sensitivity analysis, we chose an optimal interrogation window size of 320x320 pixels and a spacing of 32 pixels, resulting in a final resolution of 6.4 m. After the production of the velocity field, we filtered erroneous values using manually chosen upper and lower thresholds for both velocity and signal-to-noise ratio (SNR) - generally between 0.8-1.1 at the lower bound and 2.8-3.5 at the upper bound.

## 2.5 Uncertainty assessment

Relative uncertainties were calculated by assessing inter-DEM variation in the elevation of the exposed bedrock on Qarassap Nunata near the calving front, assuming no expected change in topography. Vertical uncertainty ($\sigma_z$) was calculated by assessing the mean per-pixel standard deviation from the mean elevation of the repeat DEMs. Horizontal uncertainty ($\sigma_{xy}$) was estimated using feature-tracked displacement fields. First, we calculate the root mean square error in displacement fields ($s_{RMSE}$) produced in the feature tracking process (Ryan et al., 2015). We assume that this value results from the combined error from the root mean square error of the two tracked images. Hence, we estimate the horizontal uncertainty ($\sigma_{xy}$) as follows:

$$\sigma_{xy} = \sqrt{\frac{s_{RMSE}^2}{2}} \tag{1}$$

Note that this estimate ignores potential error contributions from feature tracking in $s_{RMSE}$, and hence likely only provides an upper bound on the horizontal uncertainty.

From $s_{RMSE}$, we can also calculate the uncertainty of any horizontal velocity field ($\sigma_v$) as follows:

$$\sigma_v = \frac{s_{RMSE}}{\Delta t} \tag{2}$$

where $\Delta t$ is the time interval of the velocity field.

As our external orientation parameters (camera positions) are distributed densely, consistently, and evenly throughout the point cloud (cf. a GCP-based network), we assume that error is spatially non-variable, and hence that uncertainties measured at the bedrock margins are representative of error across the SfM-MVS product.

## 3 Results

### 3.1 Calving front

#### 3.1.1 DEMs and velocity fields

The calving front of Store Glacier was surveyed ten times between 10[th]-14[th] July 2017 (Table S1). Typical UAV-derived glaciological products for the calving front are shown in Figure 3, including orthophoto, DEM, and velocity field.

Our method reproduces both small- and large-scale aspects of the calving front in fine detail. At glacier-wide scales, the side of the calving front is known to have a prominent surface depression, an expression of a retreated grounding line and section of the front at floatation (e.g. Ryan et al., 2015; Todd et al., 2018). This is captured well by the GNSS-AT photogrammetry (Figure 3b; 4a). At local scales this method is accurate enough to capture the opening of crevasses over periods of days (Figure 4b; 5d–e), although reconstruction of crevasse depth continues to be problematic due to low illumination and inefficient sensor orientation within crevasses (Ryan et al., 2015).

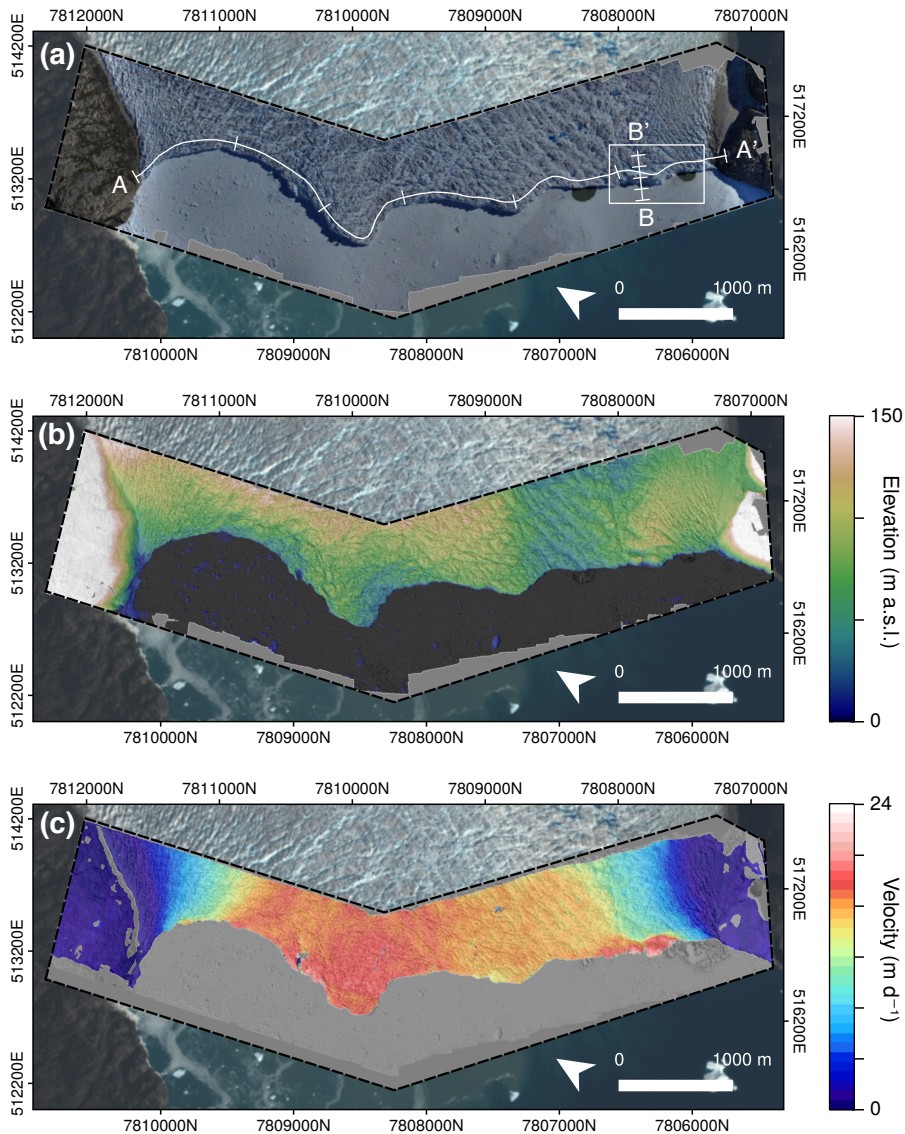

**Figure 3.** Example data output from calving front. (a) 0.15 m orthophoto, collected 10:15 12$^{th}$ July 2017. (b) 0.2 m DEM from same flight. (c) 6-hour separation velocity field ($\sigma_v = \pm 0.69$ m) from 16:15–22:15 on the 12$^{th}$ July. Transects in (a) refer to Figure 4. Box refers to location of Figure 5

The GNSS-AT method can also be successfully used to derive velocity fields of the calving front at high resolution and accuracy (Fig. 3b; 4a). The velocity field, derived from displacements detected over a six-hour period between 16:15 and 22:15 on the 12$^{th}$ July 2017 ($\sigma_v = 0.69$ m d$^{-1}$), shows that velocities are generally uniform (15 m d$^{-1}$) across much of the central calving front, with localised peaks of 20 m d$^{-1}$ (Fig. 3c; 4a; 5h). Our method is sensitive to dynamic changes at the calving

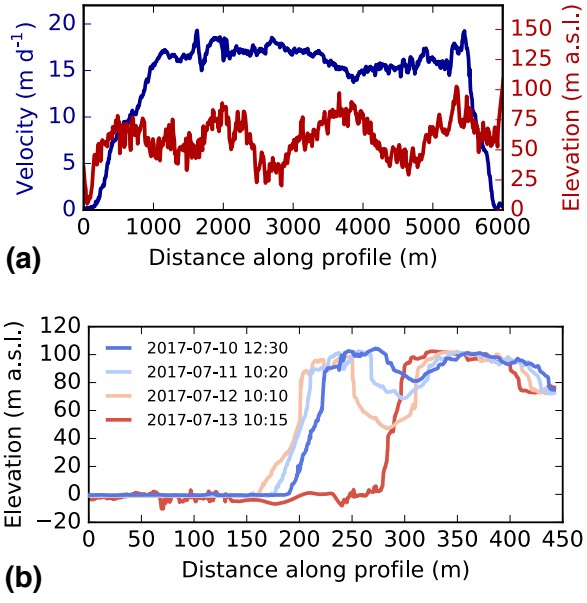

**Figure 4.** (a) Transect of A-A' in Figure 3(a), displaying velocity (blue) and elevation (red). (b) Transect of B-B' in Figure 3(a), displaying elevation profiles of the calving front through the study period. A large-scale calving event occurs between the 12[th] and 13[th] of July.

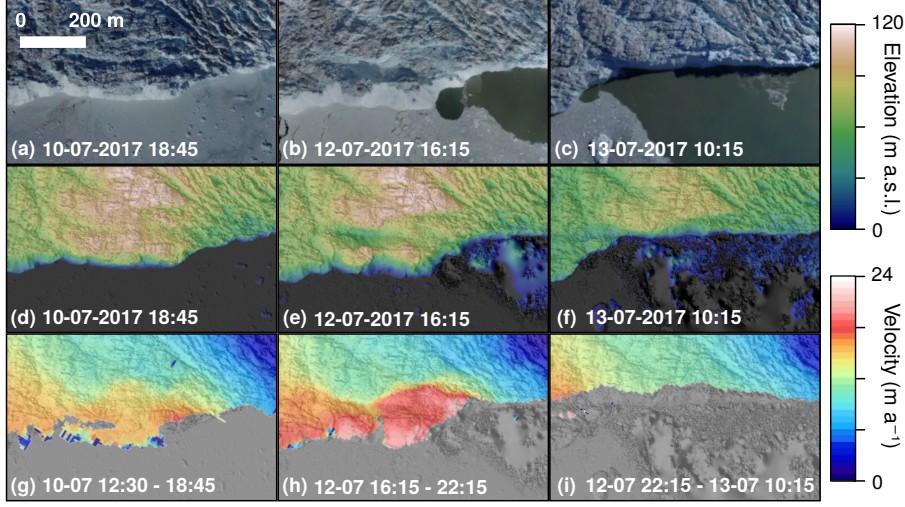

**Figure 5.** Orthophotos (a–c), DEMs (d–f) and velocity fields (e–i) showing the lead-up and aftermath of a calving event that occurred on the south side of Store Glacier between 22:15 on the 12[th] July and 10:15 on the 13[th] July. Location is marked by outline in Figure 3a. NB. the poor reconstruction of open water visible in DEMs and hillshades of figures e, f, h, and i.

front: in particular, the areas of highest velocity at the very lip of the calving front – such as regions ∼1.8–2 km and ∼5.2–5.4 km along profile A (Fig. 4) – all mark areas of ice that undergo calving events in the next 24–48 hours. One particular calving event, occurring between 22:15 on the 12$^{th}$ July and 10:15 on the 13$^{th}$ July on the southern side of Store Glacier, is detailed in Figures 4b and 5. The calving zone, measuring ∼150 000 m$^2$, occurs in a region of high shear strain in a region bordering the floating section of Store. More than 48 hours before calving, deformation in the calving zone is anomalous relative to the surrounding area: up to 20 m d$^{-1}$, whilst the region immediately behind the zone is <10 m d$^{-1}$ (Fig. 5g). Over the following two days, a plume becomes visible in front of the calving zone, opening up a region of open water in the ice melange (Fig. 5a–b). The crevasse becomes deeper and wider during this time (Fig. 4b): across transect B-B' (Fig. 3a), the crevasse increases from ∼57.5 m wide and ∼19.2 m deep at the first observation (2017-07-10 12:30) to ∼73.5 m wide and 49.6 m deep at the final observation before calving (2017-07-12 22:20). The crevasse is not obviously water filled in this period (Fig. 5b), but is filled with ice debris that has dry calved off the interior face of the crevasses (Fig. 4b; 5b), so the depths are reported above are underestimated, and water may exist beneath the debris. In the hours prior to calving, the calving zone reaches deformation rates up to 24 m d$^{-1}$ (Fig. 5h). The calving event itself resulted in the loss of an ice section ∼400 m in length extending ∼100 from the front of the glacier. Assuming the calving front is at floatation in this region of the glacier front (Todd et al., 2018), we estimate from elevation models that the total ice lost to be 9.5 x 10$^7$ m$^3$ (∼0.1 km$^3$). After calving, displacement rates at the glacier fronts return to levels consistent with the surrounding area (Fig. 5i).

### 3.1.2 Uncertainty analysis

To estimate the error of the technique, we sampled a total of 0.1 km$^2$ of bedrock across two zones close to the glacier margin where reconstruction quality matched that of the glacier surface across all DEMs (Fig. 6a). We selected eight DEMs and eight displacement fields of these sample areas, produced by feature tracking consecutive hillshaded DEMs.

The uncertainties derived from assessment of these DEMs is $\sigma_z = \pm 0.14$ m and $\sigma_{xy} = \pm 0.12$ m. This amounts to ∼1.1 times the GSD (∼11 cm) in the horizontal and ∼1.3 GSD in the vertical. The per-pixel standard deviation in the vertical axis (Fig. 6b–c) shows that vertical deviation is relatively consistent across the image. The areas of highest deviation (visible as bright yellow-white bands in Figure 6c) are pixels that are located at steep topographic cliffs, where slopes are close to vertical and thus any horizontal error will compound the reported deviation in the vertical axis. The per-pixel standard deviations in the horizontal (Fig 6d–e) reveal clustered 'hotspot' regions of high variation. However, close inspection of individual displacement fields shows that these hotspots are an artefact of individual anomalies in the displacement fields, and that areas of high deviation are not spatially consistent between displacement fields. Hence, although horizontal uncertainty is spatially variable, the variability is not dependent on factors such as surface texture or roughness, which would invalidate the assumption that a single uncertainty value can be assigned uniformly to an entire DEM.

With a displacement uncertainty $s_{RMSE} = 0.17$ m (∼1.5 GSD) and the ability to capture ice flow accurately, even along the relatively slow moving (1-5 m d$^{-1}$) sides of the glacier calving front, these uncertainty tests validate our ability to use GNSS-AT derived UAV-photogrammetry to produce accurate DEMs and velocity fields of the ice sheet interior, where there are no exposures of bedrock and ice flow is considerably slower. Because feature tracking was able to successfully track displacements

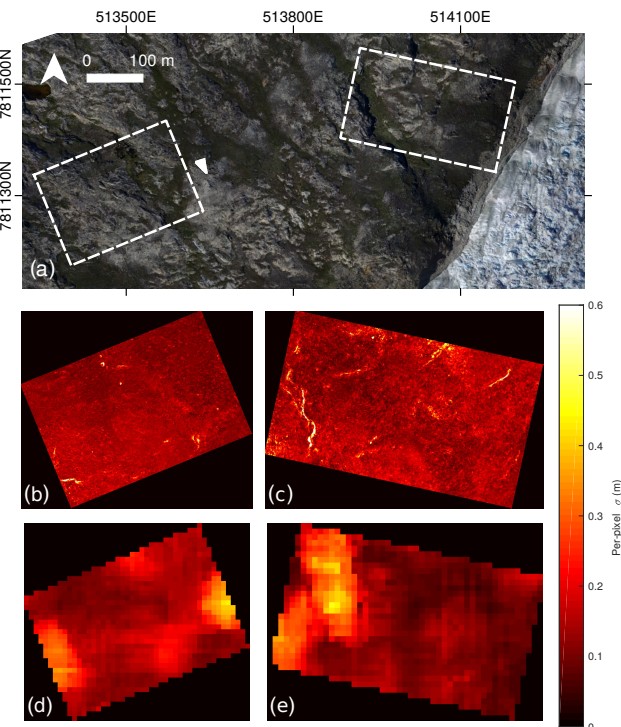

**Figure 6.** Error assessment at the calving front: (a) location of two validation sites at the calving front shown on UAV-derived orthophoto; (b-c) standard deviation in Z axis derived from DEMs (d-e) standard deviation in XY axes derived from horizontal displacement fields.

of <5 pixels, the same hardware and methodological approach should be sufficient to identify daily displacements at inland sites up to ∼60 km from the calving front of Store Glacier (i.e. to the 1 m d$^{-1}$ velocity contour).

## 3.2 Ice sheet interior

The interior study area is located 30 km inland from the calving front, where ice flows at a speed of 2 m d$^{-1}$. The location of the flight paths was motivated by the presence of a large subglacial trough identified in BedMachine v3 data, and the presence of two supraglacial lakes 28 and 31 km inland (Lake 028 and Lake 031 – see Fig. 1c). Typical UAV-derived glaciological products for the ice sheet interior (flight zone 'DT' – see Figure 1c for location) are shown in Figure 7, including orthophoto, DEM, and velocity field.

Although flight zone DT was designed to capture Lake 028, it is apparent from orthoimagery that the lake had drained prior to the beginning of the study (Fig. 7). Sentinel-2 imagery shows the drainage to occur between the 19$^{th}$ June and the 7$^{th}$ July, although Lake 031 remained filled during the study period. Lake 031 overflows into a supraglacial stream which terminates in a large (>10 m diameter) moulin formed from the hydrofracture of Lake 028. This distinct hydrological network is visible in the former lake bed (Fig. 7a), which is clearly seen as a depression in the surface DEM produced by SfM-MVS (Fig. 7b).

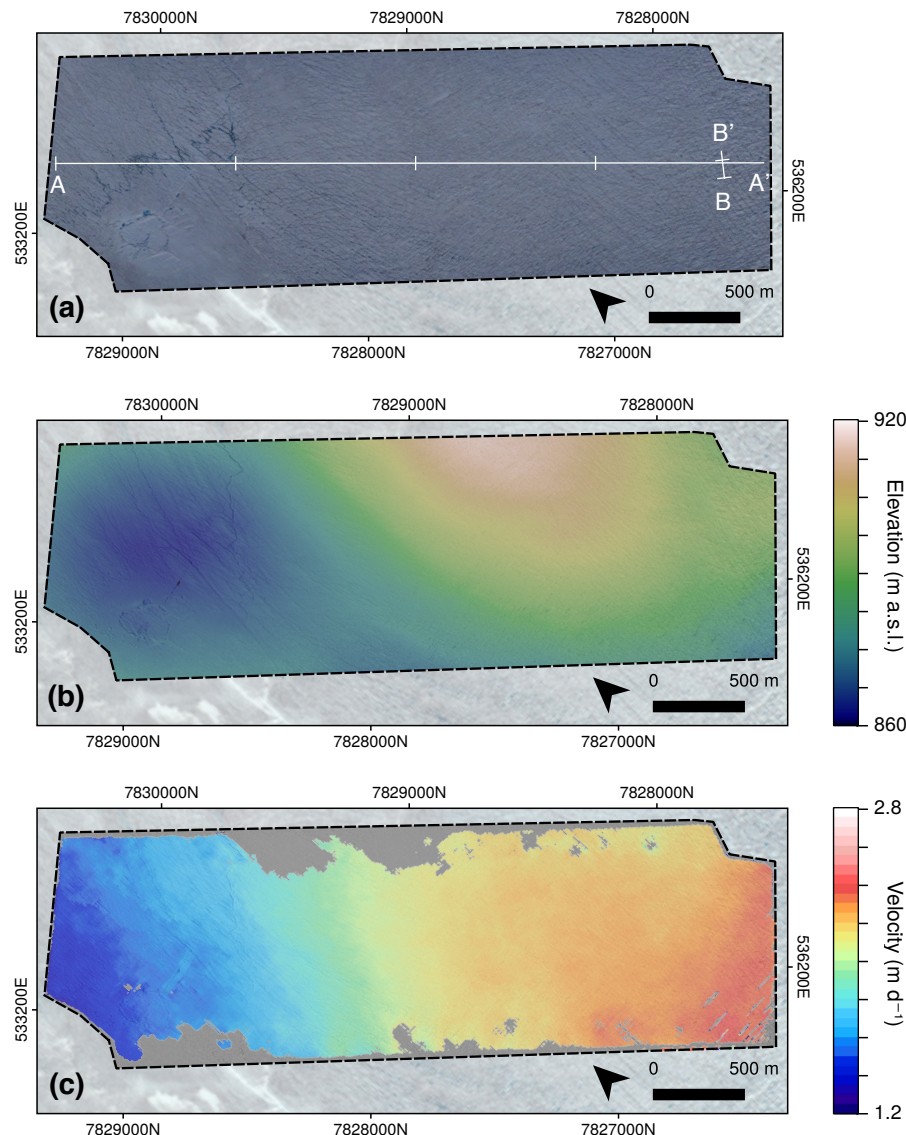

**Figure 7.** Example data output from the ice sheet interior. (a) 0.15 m orthophoto collected 15:15 22$^{nd}$ July 2017. (b) 0.2 m DEM from the same flight. (c) Velocity field ($\sigma_v = \pm 0.05$ m) from 15:15 22$^{nd}$ – 19:30 26$^{th}$ July 2017

Figures 7a–b capture two historical features of lake drainage. The first is the fracture and moulin from the 2017 lake drainage, as already described. The second is the remnant lake ice from the 2016 lake, which did not drain and is still visible as a lighter patch of ice in the western corner of Figure 7a.

Figure 7c shows a velocity field derived by feature tracking displacements on two DEM hillshades produced from orthophotos with four days seperation, from 22$^{nd}$ – 26$^{th}$ July 2017 ($\sigma_v = 0.05$ m d$^{-1}$). To our knowledge, this represents the first published

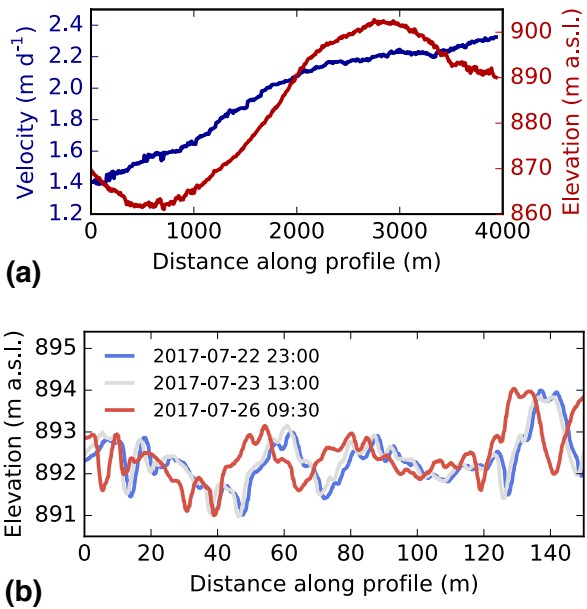

**(a)**

**(b)**

**Figure 8.** (a) Transect of A-A' in Figure 7(a), displaying velocity (blue) and elevation (red). (b) Transect of B-B' in Figure 7(a), displaying elevation profiles of a crevasse field through the study period.

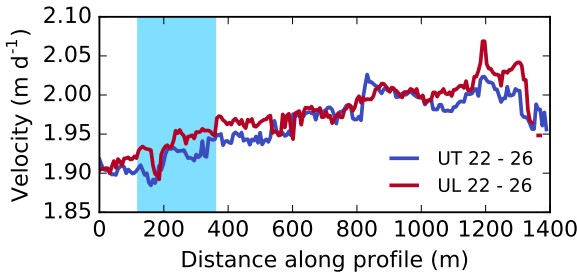

**Figure 9.** Comparison between two velocity fields obtained from different flight paths at comparable times between the 22$^{nd}$ and 26$^{th}$ July 2017. Blue shading marks where the transect crosses Lake 031. Transect is visualised as transect F9 in Figure 1c.

UAV-derived velocity field of an ice sheet interior, constructed without the use of GCPs. Feature-tracking has successfully reconstructed the full range of velocities across the interior region in which ice flow gradually increases from ∼1.4 m d$^{-1}$ in the west to ∼2.4 m d$^{-1}$ in the east (Fig. 8a). We interpret this difference to occur due to differences in bedrock topography: to the southeast, ice is flowing over a bedrock rise, the peak of which is centred approximately 2 km southeast of the study region (Fig. 1). This change in dynamics is expressed in the ice surface as an increasing frequency of deep and open crevasses (Fig. 7a; 8b).

We can test the robustness of the inland surveys by comparing contemporaneous velocity fields from independent flights. Figure 9 shows a 1.4 km velocity profile of two velocity fields, constructed for the same time period ($22^{nd}$ – $26^{th}$ June) but from two different flight paths (paths UT and UL in Fig. 1c). Despite being derived from entirely different datasets, the velocity products show remarkable agreement as they clearly fall within our estimated $\sigma_v$ uncertainty of $\pm0.05$ m d$^{-1}$ (section 2.5). Hence, cross-comparison of different datasets appears to show that velocity products are robust between varying SfM-MVS input data. Additionally, the velocity products appear to be consistent even when tracking features through water, when tracking through Lake 031 (Figure 10). Thus, although refraction at the water surface influences SfM photogrammetry in the z-axis without corrective measures (e.g. Mulsow et al., 2018), these data suggest that the horizontal accuracy of bathymetry generated by SfM-MVS photogrammetry is sufficient to measure horizontal displacement through (non-turbid) water such as supraglacial lakes.

## 4   Discussion

### 4.1   Comparison with prior methods

In this study we have estimated that, in a glacial environment flying at $\sim450$ m above ground level, SfM-MVS photogrammetric products supported by GNSS-AT geolocation can be accurate to $\pm0.12$ m ($\sim1.1$ GSD) and $\pm0.14$ m ($\sim1.3$ GSD) in the horizontal and vertical respectively. With well-constrained geolocation, horizontal (vertical) accuracies can be as high as 0.5-1.0 GSD (1.5-2.0 GSD) (Benassi et al., 2017). Our estimated accuracies are very close to these theoretical values – in fact, our vertical accuracy is even higher. Our lower accuracy in the horizontal is likely due to the fact that s$_{RMSE}$ includes feature tracking error. Common estimates of the latter can be up to 0.5 pixels (e.g. Quincey et al., 2015), which if assumed in our use case would bring the $\sigma_{xy}$ estimate down as little as $\sim0.5$ GSD. Our estimated accuracy values agree well with previously reported GNSS-AT derived estimates. For instance, Fazeli et al. (2016) report horizontal (vertical) accuracies of 0.6 (1.0) GSD using a low-cost customised multirotor UAV. Our accuracies also align with reported horizontal (vertical) accuracies of commercial fixed-wing drones, which offer similar performance to our low-cost alternative at a considerably higher price. Studies using the eBee RTK have reported horizontal (vertical) accuracies of 1.0 (1.2) GSD (Roze et al., 2014), and 0.6-1.2 (0.8-4.0) GSD (Benassi et al., 2017), and 0.8 (1.8) GSD (van der Sluijs et al., 2018), whilst the WingtraOne PPK has reported horizontal (vertical) accuracies of 1.3 (2.3) GSD (Ng and Buchheim, 2018). As a result of this level of accuracy, feature tracking can be used to reliably resolve decimetre-scale displacement (s$_{RMSE}$ = 0.17 m, $\sim1.5$ GSD) in the ice surface without the use of GCPs. For the investigation of glacier dynamics, where installing and surveying GCPs is logistically demanding, GNSS-AT therefore represents an especially significant technical advance. The method reported here can be directly compared to analogous UAV studies of Greenland glacier dynamics where both methods and uncertainty assessments have been rigorously reported.

The first example is that of Ryan et al. (2015) for Store Glacier, who were amongst the first to use UAVs in a study of Greenland Ice Sheet dynamics. Ryan et al. (2015) geolocate imagery in a two-stage procedure. First, external calibration in the SfM-MVS process was performed camera coordinates provided by an on-board autopilot navigational GPS reciever, which had an accuracy $\pm5$ m. Flying at $\sim500$ m a.s.l. (GSD $\sim0.18$ cm) provided a DEM with relative errors up to $\pm17.12$ m horizontally

($\sim$95.5 GSD) and $\pm$11.38 m vertically ($\sim$63.2 GSD), with notable warping in sea-level. A secondary stage of processing used a single GCP at the glacier margin, 3D co-registration of DEMs using visible common control points such as boulders and promontories, as well as a number of sea level control points given nominal values of zero m a.s.l.. These secondary step reduced measured RMSE across bedrock margins to $\pm$1.41 m horizontally ($\sim$7.8 GSD) and $\pm$ 1.90 m vertically ($\sim$10.6 GSD). Hence, Ryan et al. (2015) show that it is possible to achieve scientifically valuable results even without strong ground control. However, an error >1 m is of limited use to assess displacement on slower-flowing glaciers, or over short time periods – indeed, the velocity fields of Ryan et al. (2015) have notable artefacts in slow-flowing (<5 m d$^{-1}$) sectors of the calving front. The GNSS-AT method shown here provides an order-of-magnitude improvement in accuracy, as well as eliminating an additional processing step. Despite this study using a shorter time interval between flights (6 hours compared to 24 hours) – and hence evaluating velocity from a smaller magnitude of displacement – the velocity fields in this study are more accurate, permitting detection of changes in the slow-flowing sections of the ice margin. The method also successfully reconstructs a flat sea level (this can be seen in the first three transects displayed in Figure 4b – the final transect is disrupted by poor reconstruction of open water). The previous failure to reconstruct a flat sea surface of constant elevation in the first processing pass of Ryan et al. (2015) is likely a 'bowing' effect from radial lens distortion error in the fixed or self-calibrated camera calibration (James and Robson, 2014), a feature that can be reduced significantly with precise georeferencing (James et al., 2017).

Further work on UAV dynamics studies of calving fronts was developed by Jouvet et al. (2017, 2018) for Bowdoin Glacier in North Greenland. They report an improved horizontal accuracy of 10-20 cm ($\sim$1.4-2.9 GSD), a value that improves on Ryan et al. (2015), and approximately double that in this study. They achieve this level of accuracy using ten GCPs on each side of the 3 km-wide calving front, as well two GCPs on the glacier surface recorded using repeat dGPS positioning, with absolute positions of on-ice GCPs during each flight linearly interpolated. They also fly at a lower altitude ($\sim$300 m a.s.l.; GSD $\sim$7 cm) than that of Ryan et al. (2015) and this study (400-500 m a.s.l.), which improves the quality of photogrammetric reconstruction whilst limiting the total area able to be assessed in a single flight (Bowdoin is $\sim$3 km across, whilst Store is $\sim$5 km). Hence, Jouvet et al. (2017, 2018) show that it is possible to work with moving on-ice GCPs to provide viable products. However, the logistical effort is still considerable, and as a result GCP density is sparse, with large distances (up to 2 km) between GCPs, which likely leads to significant errors at points far from GCP location (Tonkin and Midgley, 2016; Gindraux et al., 2017). Additionally, linearly interpolating moving GCPs on the calving front: (i) requires that the calving front is a safe space to operate in logistically; and (ii) assumes that the glacier is moving at a constant velocity, which is a non-optimal assumption especially when studying glacier dynamics. The GNSS-AT approach applied here allows for the ability to resolve decimetre-scale displacements without depending on a GCP network. This resolves large logistical challenges at marine-terminating calving fronts, where collecting GCPs on both sides of the calving front and on the ice itself would likely require a safe operating environment, considerable time investment, and/or helicopter access, all of which are downsides that UAVs are in some way meant to alleviate.

Whilst the method described here greatly reduces the logistical requirements of working with a network of GCPs, it does not ultimately change the nature or limitations of the SfM-MVS process. For instance, the identification of key points or

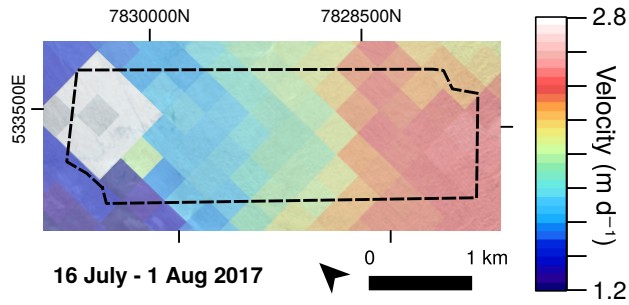

**Figure 10.** Landsat 8 OLI-derived velocity field of the study area between 16[th] July and 1[st] August 2017. Data is from GoLIVE project (Fahnestock et al., 2016, resolution = 300 m), overlaid onto Sentinel-2 optical imagery. Black outline marks the extent of the study zone in Fig. 7. Note that feature tracking has failed over the site of the former lake bed.

common features during the 3D reconstruction process will still struggle to reconstruct low texture environments such as fresh snow (Gindraux et al., 2017) or open water (visible in Figure 5e–f and the final transect of Figure 4b), as well as the true depth of crevasses (Ryan et al., 2015). Image collection should still be conducted according to best practice, including careful consideration of image overlap and flight geometry (James and Robson, 2014).

## 4.2 Applications

The case studies of a calving front and ice sheet interior provided in this study show two different applications of the GNSS-AT method: one in a calving front environment where UAVs have previously been used, and one in an ice sheet interior, where UAVs have not to date been used to assess ice dynamics. The first case study highlights that existing observations of, for instance, calving events (Ryan et al., 2015; Jouvet et al., 2017, 2018) can be successfully replicated using GNSS-AT methods (Fig. 5). However, the second case study, deriving surface velocity of an ice sheet interior, has not previously been possible using UAV-SfM methods. Instead, UAV-based ice sheet studies have largely focused on non-dynamic aspects of surface glaciology, such as albedo (e.g. Ryan et al., 2017; Burkhart et al., 2017).

Inland, opportunities for measurement of ice velocity are currently restricted to either: high-resolution in-situ GNSS measurements (e.g Doyle et al., 2015), which can capture ice velocity at extremely high temporal resolution and accuracy but only for point measurements; or satellite remote sensing techniques (e.g. Tedstone et al., 2015), which can offer regional coverage at the expense of spatial and temporal resolution (and often an inherent trade-off between the two). The opportunity for broad spatial coverage of ice velocity at high temporal resolutions (e.g. daily) is extremely limited, and often restricted to opportunistic or targeted observations where repeat intervals occur at adequate frequencies (e.g. Palmer et al., 2011; Minchew et al., 2017). UAV-based techniques allow for high-resolution velocity fields to be obtained by field researchers in targeted areas without dependency on high temporal resolution satellite observations, and with a much higher quality product than that available from global datasets and products. This quality improvement is apparent when we compare the inland velocity product in this study to a GoLIVE (Landsat-8 derived) product (Fig. 10; Fahnestock et al., 2016). While the satellite-derived data captures the overall

variation of ice flow in the study region, the acceleration from west to east is considerably less detailed. The reduced temporal resolution (16 days) results in a failure of the feature tracking algorithm to capture changes over the former lake bed, where changes in the supraglacial hydrological network disrupted feature tracking (Fig. 10). The ability to create field-based velocity fields provides new opportunities to study the spatial variation in short-term (daily-weekly) velocity variations on ice sheets,
such as those provided by supraglacial lake drainages, or variation in moulin inputs in response to rainfall or melt events.

## 4.3   Future directions

Although our method shows an improvement in accuracy relative to prior glaciological studies, this is in part due to the sub-optimal GCP placement of prior studies that is a necessary by-product of working in glacial environments where access is restricted in many places. When optimally arranged, Benassi et al. (2017) show that a dense network of ground control points
still provides a better accuracy than GNSS-AT methods, particularly vertically ($\sim$30% improvement in the horizontal and $\sim$60% in the vertical). However, it has been shown that the error of a GNSS-AT-derived product can be further constrained by the reintroduction of at least one GCP, with a final vertical accuracy only slightly worse than traditional GCP networks (Benassi et al., 2017). Whilst constructing a comprehensive network of GCPs might be difficult on glacial terrain, the introduction of one GCP, either in the form of an existing continuous GPS station, or a single target measured on a per-flight basis or interpolated
linearly as per Jouvet et al. (2017, 2018), is far more achievable than a large, dense network of GCPs. The method as described here also lacks the incorporation of directional data in the camera coordinate positions, and hence is referred to as GNSS-AT rather than full DSO (section 2.3). The navigational IMU on-board the autopilot was not precise nor accurate enough with regards to time tagging to allow full DSO. The introduction of a more precise IMU – analogous to the improvement in precision between SPS and PPK geolocation in this study – would allow full DSO geolocation in the SfM-MVS process (Cucci et al.,
2017) using a low-cost UAV system. However, we are not aware of any applied use of UAV-based DSO in the geosciences at the time of writing.

The UAV system and payload used in this study can be constructed for under £1500, which means our core hardware pushes the boundary of UAV applications in polar and other extreme environments whilst conforming to the low-cost ethos of much geoscientific UAV work. However, the full method we have described here deviates from that ethos by virtue of the dependence
on dual-frequency carrier-phase GNSS base station receivers for the differential processing of GPS data, which can often have high costs. Dual-frequency recievers are necessary for carrier phase GNSS correction over distances > 10 km, and hence as long as the UAV is equipped with a single-frequency receiver, there is a necessity for a local (< 10 km) base station to be running in parallel during the flight period. Fortunately, there has been a recent availability of cost-efficient (< USD1000) dual-frequency receivers such as the Piksi Multi (https://www.swiftnav.com/), the Tersus BX305 and BX316R (https://www.tersus-gnss.com/),
and the ComNav K501G and K708 (http://www.comnavtech.com/). These receivers present three potential innovations on the method presented here. Firstly, the integration of these low-cost systems would reduce the initial capital cost of projects that do not already have access to an expensive dual-frequency GPS receiver. Second, these GPS receivers are small and light enough to fit on small-sized UAV airframes, and hence allow for (i) an improved flight baseline and accuracy, and (ii) direct kinematic correction against an off-ice reference station (i.e. the removal of the need for processing the intermediate base station 'B1'

in ice sheet environments). Finally, these receivers could act as affordable on-ice base stations that could be distributed with a high enough density to act as low-cost 'continuous' on-ice GCPs, allowing for reduced error (as above) and validation of the final point cloud output.

## 5 Conclusions

We have presented the application of an alternative SfM-MVS geolocation method known as GNSS-supported aerial triangulation, which uses an on-board carrier-phase GNSS receiver to geolocate SfM-MVS point clouds while significantly reducing the need for GCPs. Using the calving ice front of a large Greenlandic outlet glacier as a test case, we have shown that uncertainties in the reconstruction of the glaciers surface can be reduced to $\pm0.12$ metres horizontally ($\sim1.1$ GSD) and $\pm0.14$ m vertically ($\sim1.3$ GSD), when flying at $\sim450$ m above ground level. These values compare favourably with those obtained in previous studies, which used networks of GCPs for geolocation. The elimination of ground control allows us to assess ice displacement at an inland site and to produce, to our knowledge, the first example of velocity fields derived from UAV methods at an ice sheet interior site.

The nature of studies of glacial environments inherently limits the ability of users to collect dense networks of GCPs. GNSS-AT will be of interest to those wishing to use UAV photogrammetry to obtain high-precision measurements in all glacial contexts, but will be of particular value for operation in the interior of larger ice masses, such as ice sheets, where operation away from exposed bedrock makes the collection of stable GCPs a nearly impossible task. This method has further applications, both within studies of the cryosphere – for example, in studies of sea ice – but more broadly in all geoscience applications where UAV operation occurs in hazardous environments.

*Data availability.* All derivative data used in this study (orthophotos, DEMs, velocity fields) are available upon request. Please contact Tom Chudley for this purpose (trc33@cam.ac.uk). Full source data will be made available in an online repository at the conclusion of the ERC RESPONDER project.

*Author contributions.* TRC and PC designed the study. TRC designed the UAV payload, built the UAVs, processed the GPS rover data, and processed and analysed the photogrammetry data. SHD advised on GPS techniques, collected and processed the GPS reference and base station data, and assisted with GPS rover processing. AA assisted with flight path design and photogrammetric processing. NS originally designed the UAV and helped to build the units used in this study. TRC wrote the paper with input from all authors.

*Competing interests.* The authors declare that they have no conflict of interest.

*Acknowledgements.* This research was funded by the European Research Council as part of the RESPONDER project under the European Union's Horizon 2020 research and innovation programme (grant agreement number 68304). TRC was supported by a Natural Environment Research Council Doctor Training Partnership Studentship (grant number: NE/L002507/1), and AA by the European Union's Horizon 2020 research and innovation programme under a Marie Skłodowska-Curie grant (grant agreement number 705215). We are very grateful to Ann Andreasen and the Uummannaq Polar Institute for their kind hospitality, to Nick Töberg, Samuel Cook, Sean Peters, and TJ Young for their assistance with UAV flights, and to Guillaume Jouvet and Oliver Wigmore for their constructive and helpful reviews.

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
