# Peer review of "High accuracy UAV photogrammetry of ice sheet dynamics with no ground control"

_The Cryosphere, 2018_

## Referee Comment (RC1) · Jouvet (Referee) · 8 Jan 2019

Unmanned Aerial Vehicles (UAV) photogrammetry has become a very valuable tool to generate high-resolution ortho-images and Digital Elevation Models (DEMs) for geoscientific studies at relative low cost. UAV are of special interest in glaciology, as UAVs can survey remote and hardly accessible glaciers. Repeat UAV surveying, SfM-MVS photogrammetry together with feature-tracking method can be used to track changes of the glacial terrain over time, as for instance due to ice flow motion, melt or calving. However, the inter-comparison of UAV-based photogrammetrical products requires very accurate geo-referencing. So far, it was mandatory to install Ground Control Points (GCPs) next to zones of interest to obtain a sufficient level of accuracy. This need of GCPs was therefore the Achille heel of UAV photogrammetry for glacier surveying

as it is difficult to install a network GCPs in such an extreme environments. Direct-georeferencing (not based on GCPs, but on camera locations) is key to overcome this issue, but this requires (at least) accurate geotagging of aerial pictures – a level of precision that can not be achieved by standard GPS (SPS).

This study solves this issue by the means of miniature, low-cost and cm-accurate differential GPS inboard the UAV. The added-value of this technique is illustrated by the first-ever velocity field derived by UAV of the interior of an ice sheet (i.e. without any immobile margin serving as reference, and without using any GCPs).

This is an important contribution as it introduces a promising and affordable technology for the surveying of glaciers, and their changes over time, with significant improvements compared to former methods. Beside this, the paper contains a lot of technical details that are necessary for replication.

I have one major comment (see below) about the uncertainty assessment – which is a central question in this paper. I also have others specific and minor comments that might help to improve the paper.

**Major comment**

- My main concern is about the uncertainty assessment in the horizontal direction. The authors claim that they get a 0.07 m horizontal accuracy. However, this number is based on the assumption that feature-tracking error contributes by 0.5 pixel of the uncertainty. It means that 0.07 is obtained by subtracting 0.10 (or estimated feature tracking error) to 0.17 of RMSE. Is it not a bit shortcoming? Assume the feature-tracking performs better than 0.5, it means that the uncertainty your are trying to estimate is larger. From your results, we can reasonably say from your result that 0.17 is the combined uncertainty of the two possible sources of error – this is already a very strong result! Yet, if one wants to clearly evaluate the contribution of each, this can only be done separately. For Feature-tracking

(or template matching), this might be difficult. Yet for SfM photogrammetry only, have you tried to perform surveys of an accessible terrain with GCPs for error assessment? This can be done anywhere, not necessarily in Greenland.

**Specific comments**

- What about giving the uncertainty in pixel rather than in meter (here you have sub-pixel accuracy)? If you were flying twice higher with twice more spaced lines, would you get the same sub-pixel accuracy (but the pixel would be twice bigger)? If yes, your result given in pixel would be more general. Of course, this only if the accuracy scales with the resolution (or equivalently the flight altitude). Additionally, you do not provide any estimate of the inaccuracy of Emlid Reach of the camera location (about 1 cm horizontally and vertically from my experience, not millimeter). Does the final inaccuracy of the DEM and orthophoto partly result from this small-but-existing error of the camera location or other uncertainties in the SfM-MVS (lens calibration, ...)? If both, can you quantity the contribution of each? This is related to my previous point: if the small but existing inaccuracy of the position given by Emlid Reach receiver plays no role (because it is overridden by other sources of errors), then the final accuracy of the SfM might broadly scales with the resolution (GSD). In that case, I think the result inaccuracy is better given in pixel.

- It would be good to add non-glaciological references in the discussion, e.g. studies that assess the accuracy of PPK UAV SfM photogrammetry. Of course, the results always depend on many settings that depend on the UAV equipment and type of mission flown (altitude, overlap, ...). Yet, numbers of recently release commercial products (Sensefly, Wingtra, ...) equipped with similar technology offer sub-pixel accuracy. How does your result uncertainty compare with other non-glaciological studies applying SfM UAV photogrammetry and GNSS-AT?

- It would be interesting to deepen the gain of performing multi-level flights and quantify the reduction of the bowling effect (l 22-26 p 4), e.g. by performing the SfM photogrammetry twice i) once with one flight level ii) once with two levels, and differentiate the 2 DEMs.

- I find the calving event you captured (l 26-30 p 8) very interesting, especially because you get a clear discontinuity in the velocity field indicating that a large iceberg is about to collapse. There is interesting material for discussion (influence of the plume, can this crevasses be propagating by hydro-fracturing, or similarity with the proceeding reported in (Jouvet, 2017, TC)?). However, your text is mostly factual, and does not discuss at all what processes might be responsible for these observations. Discussing these observations (in the discussion) and making a parallel with similar ones reported in the literature would be a true added-value.

- It would be more logical to move section 4.3 (Future direction) after the conclusion.

**Minor comments**

l 15 p 2 "0.09 m per m$^{-1}$" should be "0.09 m per m"?

l 35 p 2 "Here, we show that ... is used to geolocate imagery acquired over a large Greenlandic outlet glacier with a fixed-wing UAV." You could keep this statement more general (removing "Greenlandic outlet" and "fixed-wing") as your method is inherent to Greenland and type of UAV.

l 8 p 4 1 Kg looks overestimated for a Sonny alpha 6000 unless the lens is really heavy?

l 8 p 4 1500 $ sounds quiet low (my owns cost more than twice when summing the costs of all components). Does it include only the cost of the frame only? and anything else you need to fly it (RC, Second Elmid Reach, ...).

l 14 p 4   "Artupilot's Mission Panner": I would remove "Artupilot's"

l 18 p 4   GSD of $\sim$ 11 cm

l 18 p 4   "Out camera" should be "Our camera"

l 20 p 4   Did you process raw images in PhotoScan? or JPEG. I understand you processed raw otherwise why recording them in raw? If not, please, clarify.

l 10-12 p 5   "... equivalent to 2-3 mm" I'm not sure to understand these numbers. Do they refer to measure of the carrier phase? the accuracy? Maybe this sentence could be a bit more explained for novices in GNSS systems.

l 15-20 p 6   The unnecessary usage of RTK can be summarized more efficiently in a single sentence: "Whilst the Emlid Reach GNSS receiver is capable of RTK, we used instead PPK positioning for simplicity as the RTK brings additional technical constrains and is not more accurate than PPK"

l 9 p 7   DEMs always have coarser resolution than ortho-images after being produced by SfM-MSV. If you have GSD of 11 cm, 15 cm is logical resolution to output orthoimage. However, is 20 cm optimal for the DEM? Do you really get that resolution?

Section 2.5   As said before, I think it is dangerous to subtract an unknown error on the feature-tracking to $s_{\mathrm{RMSE}}$. I'd simply use $s_{\mathrm{RMSE}}$ as an upper bound, mentioning that this estimate might be sub-optimal as it might contain a contribution external to SfM.

l 16 p 8   I can see crevasses on Fig 4b, but I can not really observe crevasse opening (i.e. positive strain rate) on this figure. Either reconsider the figure or rephrase.

l 20 p 8   Fig. 3c, 4a and 5

| | |
|---|---|
| l 22 p 8 | "Our method is sensitive to small changes occurring at the calving front": I don't understand this sentence. Is the method or the calving front sensitive to small change? |
| 4, caption | A **large-scale** calving event occurs between ... of July. |
| l 14 p 11 | "track displacement of <1m" would be better given in pixel. |
| l 4-6 p 13 | This sentence seems redundant with one sentence of the section before. Maybe this is a way to avoid this redundancy. |
| 9, caption | Specify along what profile these velocity fields were taken. |
| 13-15 p 14 | It might be good to add that it worked as water of supra-glacial lake was clear/transparent. Of course, this would not work with turbid water (unlikely to be met on supra-glacial lake). |
| 16-19 p 15 | "radial error", radial with respect to what? the lens? "can be reduced significantly with the introduction of accurate aerial georeferencing (James et al., 2017)" ⇒ "can be reduced significantly with the introduction of GNSS-AT (James et al., 2017)" if this is the case, better use always the same terminology for clarity. |
| l 22 p 15 | We actually had more than 2 GCPs on each side, but instead about 10 GCPs on each side, plus a few moving with the ice. |
| Fig 2b | is too small, would require either more explanations or can be removed as well as the chain of GPS is sufficiently well explained in the text. |
| 10, caption | split last word 'lakebed' |
| 18-20 p 15 | It would be interesting to quantify the reduction of the bowing effect. |
| l 15 p 16 | high-resolution **in-situ** GNSS measurements |

l 5 p 17 "had negated cross-correlation 10" ???

l 7 p 17 velocity variation due to lake drainage or precipitation events are not neces-
sarily expected right under the suprglacial lake, but could occur far away (de-
pendent on the un/efficiency of the subglacial hydrological system). I'm won-
dering whether there is a confusion between 2 different things: i) mapping of
supraglacial lake (which might be useful to track possible drainage) ii) conse-
quence of lake drainage on the ice velocity. But the two are most likely not to be
observed in the same region.

l 21 p 17 Is full DSO geolocation already available at affordable price, or very expensive,
or still in research development? What gain would be expected using full DSO
(compared to GNSS-AT)? Could you add this information?

32-33 p 17 "Firstly, the integration of these systems allows for the realisation of a truly low-
cost, GCP-free UAV-MVS workflow for glaciological applications." This sentence
looks unnecessary to me (and even somewhat harming your work). If I under-
stand well, the gain of using dual-frequency GNSS receiver (instead of single)
is to obtain **absolutely** accurate locations, and not being constrained by the 10
km max distance separation between rover rand base? Therefore, this would re-
lax the constrain of the chain GNSS receiver (to determine the absolute position
of the Emlid Reach base station) you implemented for getting accurate absolute
georeferencing. I don't think you expect an improved relative geo-location with
dual-frequency receiver, and then a reduced uncertainty?

otion, Fig 3 'seperation' should be 'separation'

---

## Referee Comment (RC2) · Wigmore (Referee) · 17 Jan 2019

The manuscript "High accuracy UAV photogrammetry of ice sheet dynamics with no ground control" outlines a method for the collection of high accuracy DEMs and orthomosaics from UAV without the use of GCPs, referred to as GNSS-AT. This method is applied to two study sites (calving face and internal region) on the Store glacier in Greenland. Perhaps the greatest limitation to UAV surveys is the collection of GCPs, this is particularly true for glacier environments where access to the survey area is often impossible. The method outlined by the authors provides a low cost, relatively simple method to solve this problem. It is important to point out that this solution is not especially novel, as it has been implemented relatively widely outside of glaciology. However, this is certainly one of (if not) the first applications of the method to

glaciology and is applied over an interesting and challenging environment, providing new insights into these systems. I believe this paper is a unique contribution and will find widespread interest within the glaciologic community, and within the broader earth sciences. The authors do an excellent job of explaining the method in a logical and easy to follow manner. The manuscript is well written and logically laid out.

Major Comments:

One issue that is not discussed is the role of UAV pitch, roll and yaw on the accuracy of the camera positions. The authors apply a standard offset of -7.9cm in Y direction and +13.2cm in the Z direction. However, the actual offset between the antenna centre and the camera sensor centre will change as a function of the attitude of the UAV itself. The only way to correct this is to collect accurate IMU measurements of the UAV/Camera attitude at time of image capture. Given the financial constraints of a high quality IMU (which are clearly discussed) this is not possible. But it would be good to include mention of this as a possible error source and to calculate a ballpark estimate of this error. This can be done by looking at the IMU record from the pixhawk to identify the range of pitch/roll/yaw experienced during the flight and then calculating the impact of these values on the offset values. Also, a picture of the UAV setup showing this offset would help. Finally, I presume that the Y offset would flip depending which direction the UAV is flying? Is that the case? And if so how is it accounted for? Best practice for this design and without an IMU is to install the GNSS antenna directly above the camera so that only the Z offset is considered, and this is more or less static given vertical offsets are typically small.

While this technique is relatively new in its application to glaciology, there have been a number of journal publications and grey literature that discuss the benefits and accuracy of GNSS-AT methods. The authors should include some of these citations and briefly mention this work in the introduction and/or discussion. Also some of these papers and industry grey literature compare GNSS-AT methods against more common GCP methods, which is arguably the best way to assess the accuracy of GNSS-AT.

This is not done in this paper so would provide strong justification for the viability of the method.

One final critique is the accuracy assessment. While I believe what was done is acceptable and does prove the method, it would be beneficial to have some sort of external comparison. Either by comparing the GNSS-AT method to the more typical (and accurate) GCP method, or to some other dataset, e.g. terrestrial/airborne LiDAR. This could be done over a small subset area where access is not as much of an issue – e.g. the bedrock zone.

Specific Comments:

3:4 technically the second 'bedrock' exposure area is feasible to install GCPs, reword this accordingly.

3:5 prohibits should be prohibit

3:12 at last 1948 should be at least 1948

3:16 missing units, guessing km – should be "at least 30 km inland"

4:7 Delete "a" – "the UAV is capable of ∼1hour of flight time at ∼60km"

4:16 "allowing flight plans to avoid collision with cliffs" this sentence is awkwardly worded, perhaps rephrase

4:18 "Out camera" should be "Our camera"

4:25 Great method! Is there a reason you didn't do this over the calving face as well?

5:15 Repeats information presented briefly at 4:10 – Perhaps you can remove the text at 4:10 as an expanded discussion is included here.

6:14 "was achievable" do you mean "was located nearby"

6:14 If you used the EMLID reach in RTK mode you would be using the EMLID as the base station, correct? In which case the base station would only be single frequency.

So this is another reason not to use the EMLID in RTK mode I believe. Or can EMLID interpret RTK corrections from other base station receivers (e.g. trimble etc?). On that note, I didn't see the make and model for the local dual frequency base station "B1". Also, reference station is trimble Net R9 receiver, but antenna not provided. Good to include these details if available.

6:15 PPK solutions are also often more accurate than RTK solutions as precise ephemeris data for the GNSS satellites is available during post processing.

8:20 detected over "a" six-hour period

8:27 More "than" 48 hours. . ..

11:5 reveals, perhaps reveal?

12:1 Couldn't you install the on ice base station for a few days (anywhere on the ice) and then process that data using the NRCAN PPP kinematic processing method (or OPUS) and then just pull the position of the base station at the time of the flight and use that as the static position for PPK processing of the UAV. That would mean you could work anywhere on the ice sheet and wouldn't be constrained to the 60km distance from the static reference station installed on bedrock.

14:4 the peak of which "is" centred

15:27 "moving one-ice GCP" should be "on-ice"

17:35 I think another point here is that dual frequency tends to provide more accurate positions on moving objects (i.e. UAV) especially in the vertical.

---

## Author Comment (AC1) · 20 Feb 2019

The comment was uploaded in the form of a supplement:
https://www.the-cryosphere-discuss.net/tc-2018-256/tc-2018-256-AC1-supplement.pdf
* * *

---

## Author Response (AR1)

Tom Chudley
Scott Polar Research Institute
University of Cambridge
Lensfield Road
Cambridge CB2 1ER
United Kingdom

Tel: +44 (0) 1223 336574
Email: trc33@cam.ac.uk

20th February 2019

Dear Dr Karlsson,

**RE: Response to Referee Comments**

We have received two reviews to our manuscript tc-2018-256 *High accuracy UAV photogrammetry of ice sheet dynamics with no ground control*, both of which were generally supportive of our work and the relevance of the discussed method to the community.

We have adapted the manuscript in light of their constructive comments, for which we are grateful. In addition, we have made very minor textual changes to the manuscript that were not directly requested by reviewers, in order to improve the clarity of the manuscript in light of the reviewer recommendations, or to fix minor errors in the text.

In our responses below, referee comments are shown in italicised **blue**, our response in **black** and changes in manuscript in **red**. We have also attached a revised manuscript with highlighted changes: page numbers in the text below refer to this highlighted version of the manuscript.

Thank you for your consideration of our revised manuscript, which we hope is now acceptable for publication. Please continue to address further correspondence to me at trc33@cam.ac.uk.

Sincerely,

Tom Chudley and Co-authors

**Comments by Reviewer #1, Guillaume Jouvet, and responses:**

*Unmanned Aerial Vehicles (UAV) photogrammetry has become a very valuable tool to generate high-resolution ortho-images and Digital Elevation Models (DEMs) for geoscientific studies at relative low cost. UAV are of special interest in glaciology, as UAVs can survey remote and hardly accessible glaciers. Repeat UAV surveying, SfM-MVS photogrammetry together with feature-tracking method can be used to track changes of the glacial terrain over time, as for instance due to ice flow motion, melt or calving. However, the inter-comparison of UAV-based photogrammetrical products requires very accurate geo-referencing. So far, it was mandatory to install Ground Control Points (GCPs) next to zones of interest to obtain a sufficient level of accuracy. This need of GCPs was therefore the Achille heel of UAV photogrammetry for glacier surveying as it is difficult to install a network GCPs in such an extreme environments. Direct georeferencing (not based on GCPs, but on camera locations) is key to overcome this issue, but this requires (at least) accurate geotagging of aerial pictures – a level of precision that can not be achieved by standard GPS (SPS).*

*This study solves this issue by the means of miniature, low-cost and cm-accurate differential GPS inboard the UAV. The added-value of this technique is illustrated by the first-ever velocity field derived by UAV of the interior of an ice sheet (i.e. without any immobile margin serving as reference, and without using any GCPs).*

*This is an important contribution as it introduces a promising and affordable technology for the surveying of glaciers, and their changes over time, with significant improvements compared to former methods. Beside this, the paper contains a lot of technical details that are necessary for replication. I have one major comment (see below) about the uncertainty assessment – which is a central question in this paper. I also have others specific and minor comments that might help to improve the paper.*

We are grateful to Dr Jouvet for his helpful and supportive review. Below, we show how we have revised the manuscript in light of his comments and recommendations.

**Major comment**

*My main concern is about the uncertainty assessment in the horizontal direction. The authors claim that they get a 0.07 m horizontal accuracy. However, this number is based on the assumption that feature-tracking error contributes by 0.5 pixel of the uncertainty. It means that 0.07 is obtained by subtracting 0.10 (or estimated feature tracking error) to 0.17 of RMSE. Is it not a bit shortcoming? Assume the feature-tracking performs better than 0.5, it means that the uncertainty your are trying to estimate is larger. From your results, we can reasonably say from your result that 0.17 is the combined uncertainty of the two possible sources of error – this is already a very strong result! Yet, if one wants to clearly evaluate the contribution of each, this can only be done separately. For Feature-tracking (or template matching), this might be difficult. Yet for SfM photogrammetry only, have you tried to perform surveys of an accessible terrain with GCPs for error assessment? This can be done anywhere, not necessarily in Greenland.*

And also specific comment: *Section 2.5 As said before, I think it is dangerous to subtract an unknown error on the featuretracking to sRMSE. I'd simply use sRMSE as an upper bound, mentioning that this estimate might be sub-optimal as it might contain a contribution external to SfM.*

We agree with this assessment. Whilst 0.5 pixels is a typical value to apply for feature-tracking uncertainty, it is likely not equally transferable to this use case. Instead, we have chosen to ignore feature tracking error and estimate an upper bound for the horizontal uncertainty as follows:

$$S_{RMSE} = \sqrt{\sigma_{xy}{}^2 + \sigma_{xy}{}^2}$$

This provides a value of 0.12 cm (1.1 GSD), which we clarify in-text includes feature-tracking error and is likely an upper limit (this is supported by comparisons to alternative studies, which normally have a horizontal error 0.5-1.0 GSD).

We have changed (i) the methods (P8L17-28), (ii) relevant references throughout the text to horizontal uncertainty (P1L10-11, P12L11, P16L4-5, P20L5-6), and (iii) added text in the discussion clarifying the inclusion of feature tracking error when comparing to other papers that have estimated GNSS-AT error (P16L8-10) (see below).

**Specific comments**

*What about giving the uncertainty in pixel rather than in meter (here you have sub-pixel accuracy)? If you were flying twice higher with twice more spaced lines, would you get the same sub-pixel accuracy (but the pixel would be twice bigger)? If yes, your result given in pixel would be more general. Of course, this only if the accuracy scales with the resolution (or equivalently the flight altitude).*

It is important that we deal with uncertainties in metres and not only in pixels as consideration of uncertainty in velocity is ultimately dependent on displacement uncertainty in metres. However, we agree that the it would be useful to present relative pixel uncertainties, not least because it would place our absolute uncertainties into context (decimetres cf. centimetres in other studies due to varying flight altitudes). We choose to present this not as pixels but as units of ground sampling distance (GSD), which is the generalised unit used in other studies. We have referred to this new uncertainty (~1.1 GSD in the horizontal and ~1.3 GSD in the vertical) alongside absolute values throughout the paper (P1L10-11, P12L11, P16L4-5, P20L5-6). Further, this value allows comparison with alternative non-glaciological studies, which does show that relative uncertainty is broadly the same (i.e. scales with altitude) once normalised against GSD - see further discussion below.

*Additionally, you do not provide any estimate of the inaccuracy of Emlid Reach of the camera location (about 1 cm horizontally and vertically from my experience, not millimeter). Does the final inaccuracy of the DEM and orthophoto partly result from this small-but-existing error of the camera location or other uncertainties in the SfM-MVS (lens calibration, ...)? If both, can you quantity the contribution of each? This is related to my previous point: if the small but existing inaccuracy of the position given by Emlid Reach receiver plays no role (because it is overridden by other sources of errors), then the final accuracy of the SfM might broadly scales with the resolution (GSD). In that case, I think the result inaccuracy is better given in pixel.*

Regarding Emlid Reach uncertainty: RTKLIB output reports standard deviations of ~0.6 mm horizontally and ~1.1 mm vertically for fixed solution data. Given the final estimated uncertainty of the order of a decimetre, we assume the Emlid Reach uncertainty is subsumed into other uncertainties - largely that of the

photogrammetry itself, that of lens calibration error, etc., that exist in the 'black box' of Photoscan. However, we don't feel in a position to quantify this, and cannot find any examples of this being done in previous papers utilising GNSS-AT, which also present a single 'cumulative' error estimate. With regards to your last point, It is worth pointing out, given the above, that this might *not* be the case at lower elevation, where ~1cm GPS error is a significant component of the 2-3cm final error.

We have explicitly included final reported GPS uncertainty in the methods section 2.3:

P7L25: RTKLIB camera location outputs are estimated to have standard deviations ~0.6 mm horizontally and ~1.1 mm vertically for fixed solution data.

*It would be good to add non-glaciological references in the discussion, e.g. studies that assess the accuracy of PPK UAV SfM photogrammetry. Of course, the results always depend on many settings that depend on the UAV equipment and type of mission flown (altitude, overlap, ...). Yet, numbers of recently release commercial products (Sensefly, Wingtra, ...) equipped with similar technology offer sub-pixel accuracy. How does your result uncertainty compare with other non-glaciological studies applying SfM UAV photogrammetry and GNSS-AT?*

Thank you for this recommendation - it has proven particularly useful as a point of comparison. We have taken note of various reported accuracies (our estimated errors match well with previous estimates) and included them as points of comparison in the discussion:

P16L5-16: With well-constrained geolocation, horizontal (vertical) accuracies can be as high as 0.5-1.0 GSD (1.5-2.0 GSD) (Benassi *et al.* 2017). Our estimated accuracies are very close to these theoretical values -- in fact, our vertical accuracy is even higher. Our lower accuracy in the horizontal is likely due to the fact that $s_{RMSE}$ Includes feature tracking error. Common estimates of the latter can be up to 0.5 pixels (e.g. Quincey et al., 2015), which if assumed in our use case would bring the $\sigma_{xy}$ estimate down as little as ~0.5 GSD. Our estimated accuracy values agree well with previously reported GNSS-AT derived estimates. For instance, Fazelli *et al.* (2016) report horizontal (vertical) accuracies of 0.6 (1.0) GSD using a low-cost customised multirotor UAV. Our accuracies also align with reported horizontal (vertical) accuracies of commercial fixed-wing drones, which offer similar performance to our low-cost alternative at a considerably higher price. Studies using the eBee RTK have reported horizontal (vertical) accuracies of 1.0 (1.2) GSD (Roze *et al.* 2014), and 0.6-1.2 (0.8-4.0) GSD (Benassi *et al.* 2017), and 0.8 (1.8) GSD (van der Sluijs *et al.* 2018), whilst the WingtraOne PPK has reported horizontal (vertical) accuracies of 1.3 (2.3) GSD (Ng and Buchheim, 2018).

Furthermore, we have expanded our range of methodological references in the introduction (we have outlined this fully in a response to similar comment by reviewer #2).

*It would be interesting to deepen the gain of performing multi-level flights and quantify the reduction of the bowling effect (l 22-26 p 4), e.g. by performing the SfM photogrammetry twice i) once with one flight level ii) once with two levels, and differentiate the 2 DEMs.*

And also minor comment: *l 18-20 p 15 It would be interesting to quantify the reduction of the bowling effect.*

In these two comments, it is suggested that it would be interested to quantify the impact of (i) oblique imagery and (ii) GNSS-AT on the established warping effect visible in non-optimal SfM surveys. Whilst we agree it would be interesting to quantify the impact, given: (i) the scope of the study; (ii) the lack of a control dataset e.g. from terrestrial laser scanning; and (iii) our relative expertise, we do not think we would be able to add anything meaningful to the literature. This topic has been given a comprehensive treatment by Mike James and colleagues in their relevant papers (James and Robson, 2014, and James *et al.* 2017 - in particular see fig. 6 of the latter) and are referenced in the text at appropriate points. However, since this work has been in discussion, another excellent detailed assessment of the impact of oblique imagery has been published by Nesbit and Hugenholtz (2019), which also makes specific practical recommendations as to the when and how of incorporation of oblique imagery into flight routines. We have added reference to this in the text (P5L10). Given that we may be the first to incorporate this recommendation in the glaciological literature, we hope these detailed assessments will provide other researchers with enough information to be able to put these techniques - particularly the relatively simple oblique image capture - into wider practice.

*I find the calving event you captured (l 26-30 p 8) very interesting, especially because you get a clear discontinuity in the velocity field indicating that a large iceberg is about to collapse. There is interesting material for discussion (influence of the plume, can this crevasses be propagating by hydro-fracturing, or similarity with the proceeding reported in (Jouvet, 2017, TC)?). However, your text is mostly factual, and does not discuss at all what processes might be responsible for these observations. Discussing these observations (in the discussion) and making a parallel with similar ones reported in the literature would be a true added-value.*

We agree that the calving event shown in the results is interesting. However, we consider that detailed analysis and discussion of the event (along with the others observed across the study period) belong as part of their own study - the event shown here is simple a case study to highlight a potential use case of the method. Nevertheless, considering that even the limited data shown here may be of interest to others working on calving dynamics, we have included additional information in the relevant results section, including quantifications of crevasse width, depth, and calving volume (P9L30-P12L5). We have added an additional range of subfigures to Figure 5 (d--f) showing DEMs that are helpful to visualise crevasse formation, and have changed the calving cross-section shown in Figure 4b so it matches that of the calving event in Figure 5, in order to improve the flow and cohesion of the results.

*It would be more logical to move section 4.3 (Future direction) after the conclusion.*

We have retained section 4.3 as it was, as we prefer to finish the manuscript with the conclusions.

**Minor Comments**

*l 15 p 2 "0.09 m per m–1" should be "0.09 m per m"?*

Corrected.

*l 35 p 2 "Here, we show that ... is used to geolocate imagery acquired over a large Greenlandic outlet glacier with a fixed-wing UAV." You could keep this statement more general (removing "Greenlandic outlet" and "fixed-wing") as your method is inherent to Greenland and type of UAV.*

Rewriting of this paragraph in response to comments by Reviewer #2 has resulted in the removal of this sentence.

*l 8 p 4 1 Kg looks overestimated for a Sonny alpha 6000 unless the lens is really heavy?*

Agreed - 1 kg refers to our estimate of the maximum potential payload given our configuration. The true scientific payload weight (including camera, trigger, L1 GPS, antenna, and ground plane) is approx. 500 g. Rewritten (P4L20-21).

*l 8 p 4 1500 $ sounds quiet low (my owns cost more than twice when summing the costs of all components). Does it include only the cost of the frame only? and anything else you need to fly it (RC, Second Elmid Reach, ...).*

NB that we refer to the cost in GBP, not USD, so the true value is closer to USD 2000 - although this does not account fully for the difference in our estimates.

We have referred back to our components list and re-checked prices and can confirm that this this is an accurate estimate for the set up of a complete standalone unit including Skywalker frame, PixHawk autopilot (and supporting hardware), flight hardware (LiPos, ESC, motor, servos, props etc.), and scientific hardware (camera, GPS, and supporting hardware). However, it does not include the ground components (Tx and LiPo charger, which came to ~GBP 300), nor the Trimble GPS ground equipment, which, as discussed in section 4.3, is of considerably higher cost, albeit often a common pre-existing part of glaciological field campaigns.

Given the above, we hope that the discrepancy is less than originally considered, and to within the bounds of reasonable error given individual hardware choices, pricing variations, and recent currency fluctuations. We have altered the text to clarify the cost refers to a single complete flight unit:

P4L25-28: The cost of a complete flight unit (including frame, hardware, and scientific payload) was approximately ~ £1500 per unit. Further necessary ground equipment, which could be shared between units, came to ~£300: this includes the radio transmitter and lithium polymer battery charger, but not the ground-based GPS (sections 2.3, 4.3).

*l 14 p 4 "Artupilot's Mission Panner": I would remove "Artupilot's"*

In the interests of describing a complete and detailed workflow, we feel it is important to refer at least once to the exact software used - particularly as this software is not synonymous with (albeit closely associated to) Arduplane firmware.

*l 18 p 4 GSD of ~ 11 cm*

Corrected.

*l 18 p 4 "Out camera" should be "Our camera"*

Corrected.

*l 20 p 4 Did you process raw images in PhotoScan? or JPEG. I understand you processed raw otherwise why recording them in raw? If not, please, clarify.*

We chose to record in RAW to avoid the potential influence of processing lossy file formats such as JPEG. Photoscan cannot process RAW images, but were converted to 16-bit Tiff format for AgiSoft (see also Ryan *et al.* 2017). We have clarified the sentence as follows:

P5L2-3: Photos were recorded in RAW format to ensure lossless storage of images, and converted into Photoscan-compatible 16-bit tiffs before processing.

*l 10-12 p 5 "... equivalent to 2-3 mm" I'm not sure to understand these numbers. Do they refer to measure of the carrier phase? the accuracy? Maybe this sentence could be a bit more explained for novices in GNSS systems.*

We have rewritten the sentence for greater clarity:

P5L27-29: Higher positioning accuracy than is offered by the SPS can be achieved by using differential carrier phase positioning, which makes use of the ability of GNSS receivers to measure the carrier phase to one hundredth of a cycle, equivalent to about 2 mm in distance (Leick *et al.* 2004)

*l 15-20 p 6 The unnecessary usage of RTK can be summarized more efficiently in a single sentence: "Whilst the Emlid Reach GNSS receiver is capable of RTK, we used instead PPK positioning for simplicity as the RTK brings additional technical constrains and is not more accurate than PPK"*

In an effort to balance this recommendation against a recommended addition from reviewer #2, we have rewritten the paragraph as follows.

P7L1-8: Whilst the Emlid Reach GNSS receiver is capable of real-time kinematic (RTK) we used instead post-processed kinematic (PPK) positioning for three primary reasons. First, PPK does not rely on maintaining a reliable real-time radio link with a GNSS base station, which would introduce additional technical constraints. Second, PPK solutions are also often more accurate than RTK solutions as precise ephemeris data for the GNSS satellites is available during post processing. Third, absolute positioning using

RTK requires a stationary reference station with a known position, which is not possible in real time on an advecting ice surface.

Above, we simplify part of the explanation, but choose to keep the additional clarification about absolute positioning using RTK, which is an additional constraint for operating on ice sheets that some readers may not be concerned with (and hence decide that RTK is viable).

*l 9 p 7 DEMs always have coarser resolution than ortho-images after being produced by SfM-MSV. If you have GSD of 11 cm, 15 cm is logical resolution to output orthoimage. However, is 20 cm optimal for the DEM? Do you really get that resolution?*

An optimal DEM resolution is recommended by Photoscan: given the proprietary / black box nature of Photoscan it is hard to assess the exact nature of this recommendation, but it is apparently based on the effective point cloud resolution (which scales with quality of reconstruction), with some additional leeway provided to avoid interpolation. We were consistently recommended a resolution of ~18 cm across point clouds, so chose 20 cm as a coarser option to accommodate some further variation. We have added the following qualifier in the text:

P8L2: ... based on recommended output resolutions from Photoscan.

*l 16 p 8 I can see crevasses on Fig 4b, but I can not really observe crevasse opening (i.e. positive strain rate) on this figure. Either reconsider the figure or rephrase.*

Tto provide an alternative example, we have replaced the transect in Figure 4b  with another (see updated Fig. 1a) where crevasse opening is clearly visible. This new transect covers the same calving event shown in Figure 5, and hence provides additional cohesion through the section. We have also referenced new subfigures 5d--e, where the opening of this crevasse in plane view is clearly visible in the DEMs of figure 5d and 5e.

*l 20 p 8 Fig. 3c, 4a and 5*

We have added references to these figures.

*l 22 p 8 "Our method is sensitive to small changes occurring at the calving front": I don't understand this sentence. Is the method or the calving front sensitive to small change?*

The method. We have rewritten as follows to clarify:

P7L22-23: Our method is sensitive to relatively small dynamic changes at the calving front

*Fig 4, caption A large-scale calving event occurs between ... of July.*

We have added 'large-scale' to the description.

*l 14 p 11 "track displacement of <1m" would be better given in pixel.*

Changed to < 5 pixels.

*l 4-6 p 13 This sentence seems redundant with one sentence of the section before. Maybe this is a way to avoid this redundancy.*

Changed as follows:

P14 L1-2: We can test the robustness of the inland surveys by comparing contemporaneous velocity fields from independent flights.

*Fig 9, caption Specify along what profile these velocity fields were taken.*

Thanks for highlighting this absence. Transect F9 has been added to fig. 1c. The caption has been edited to reflect this.

*l 13-15 p 14 It might be good to add that it worked as water of supra-glacial lake was clear/transparent. Of course, this would not work with turbid water (unlikely to be met on supra-glacial lake).*

Agreed. Changed as follows:

P15L6-7: ...is sufficient to measure horizontal displacement through (non-turbid) water such as supraglacial lakes.

*l 16-19 p 15 "radial error", radial with respect to what? the lens? "can be reduced significantly with the introduction of accurate aerial georeferencing (James et al., 2017)" ⇒ "can be reduced significantly with the introduction of GNSS-AT (James et al., 2017)" if this is the case, better use always the same terminology for clarity.*

Yes, the lens. Sentence rewritten as follows:

P17L3-6: ...likely a 'bowing' effect from radial lens distortion error in the fixed or self-calibrated camera calibration (James and Robson, 2014), a feature that can be reduced significantly with the introduction of precise georeferencing (James et al., 2017).

*l 22 p 15 We actually had more than 2 GCPs on each side, but instead about 10 GCPs on each side, plus a few moving with the ice.*

Thanks for clarification. We have corrected the GCP counts in the text (P17L9).

*Fig 2b is too small, would require either more explanations or can be removed as well as the chain of GPS is sufficiently well explained in the text.*

We consider figure 2b to be a useful visual aid to support description in the text so have chosen not to remove it - however, we have altered and enlarged the subfigure to better fit the aspect ratio provided by figure 2.

*Fig 10, caption split last word 'lakebed'*

Corrected.

*l 15 p 16 high-resolution in-situ GNSS measurements*

Corrected.

*l 5 p 17 "had negated cross-correlation 10" ???*

Corrected (and clarified).

P18L16: ...where changes in the supraglacial hydrological network disrupted feature tracking (Fig. 10).

*l 7 p 17 velocity variation due to lake drainage or precipitation events are not necessarily expected right under the supraglacial lake, but could occur far away (dependent on the un/efficiency of the subglacial hydrological system). I'm wondering whether there is a confusion between 2 different things: i) mapping of supraglacial lake (which might be useful to track possible drainage) ii) consequence of lake drainage on the ice velocity. But the two are most likely not to be observed in the same region.*

Supraglacial lake drainage results in local accelerations in ice velocity that have previously been observed with in-situ GNSS measurements (Doyle *et al.* 2013, among others). UAV-derived data would be able to show both the geomorphological changes referred to in point (i) and the impacts on ice velocity referred to in point (ii). We have real examples of both from our 2018 field season - this will be published at a later date.

*l 21 p 17 Is full DSO geolocation already available at affordable price, or very expensive, or still in research development? What gain would be expected using full DSO (compared to GNSS-AT)? Could you add this information?*

DSO is well established for aerial photogrammetry, and has been proven for UAV applications (see Cucci *et al.* 2017, which is also cited in-text). However, we are not aware of any applied geoscience that utilises full DSO. Speculating, this may be because 'we' (geoscience teams looking to apply existing, if novel, methods

rather than develop new ones) are already familiar with GPS post-processing from other field applications, and are more comfortable with applying this in UAV scenarios. With regards to potential pricing, the Piksi Multi (the $1000 dual-frequency example) comes with an IMU, and hence should theoretically allow for full ISO, but at the time of writing their white paper appears out-of-date and does not utilise the IMU, instead relying on the IMU in the PixHawk autopilot module, which introduces problems due to the low data quality and temporal matching.

To summarise this digression - we have added the following sentence at the paragraph in question:

P19L12-13: However, we are not aware of any applied use of UAV-based DSO in the geosciences at the time of writing.

*I 32-33 p 17 "Firstly, the integration of these systems allows for the realisation of a truly lowcost, GCP-free UAV-MVS workflow for glaciological applications." This sentence looks unnecessary to me (and even somewhat harming your work). If I understand well, the gain of using dual-frequency GNSS receiver (instead of single) is to obtain absolutely accurate locations, and not being constrained by the 10 km max distance separation between rover and base? Therefore, this would relax the constraint of the chain GNSS receiver (to  determine the absolute position of the Emlid Reach base station) you implemented for getting accurate absolute georeferencing. I don't think you expect an improved relative geo-location with dual-frequency receiver, and then a reduced uncertainty?*

Thank you for the useful comment. We have rewritten the first two points as follows, which hopefully more clearly highlights the key advantages of (i) cost and (ii) baseline extension.

P19L23-30: Firstly, the integration of these low-cost systems would reduce the initial capital cost of projects that do not already have access to an expensive dual-frequency GPS receiver. Second, these GPS receivers are small and light enough to fit on small-sized UAV airframes, and hence allow for (i) an extended flight baseline, and (ii) direct kinematic correction against an off-ice reference station (i.e. the removal of the need for processing the intermediate base station `B1' in ice sheet environments)

*caption, Fig 3 'seperation' should be 'separation'*

Corrected.

**Comments by Reviewer #2, Oliver Wigmore, and responses:**

*The manuscript "High accuracy UAV photogrammetry of ice sheet dynamics with no ground control" outlines a method for the collection of high accuracy DEMs and orthomosaics from UAV without the use of GCPs, referred to as GNSS-AT. This method is applied to two study sites (calving face and internal region) on the Store glacier in Greenland. Perhaps the greatest limitation to UAV surveys is the collection of GCPs, this is particularly true for glacier environments where access to the survey area is often impossible. The method outlined by the authors provides a low cost, relatively simple method to solve this problem. It is important to point out that this solution is not especially novel, as it has been implemented relatively widely outside of glaciology. However, this is certainly one of (if not) the first applications of the method to glaciology and is applied over an interesting and challenging environment, providing new insights into these systems. I believe this paper is a unique contribution and will find widespread interest within the glaciologic community, and within the broader earth sciences. The authors do an excellent job of explaining the method in a logical and easy to follow manner. The manuscript is well written and logically laid out.*

We are grateful to Dr Wigmore for his helpful and supportive review. Below, we show how we have revised the manuscript in light of his comments and recommendations.

**Major Comments**

*One issue that is not discussed is the role of UAV pitch, roll and yaw on the accuracy of the camera positions. The authors apply a standard offset of -7.9cm in Y direction and +13.2cm in the Z direction. However, the actual offset between the antenna centre and the camera sensor centre will change as a function of the attitude of the UAV itself. The only way to correct this is to collect accurate IMU measurements of the UAV/Camera attitude at time of image capture. Given the financial constraints of a high quality IMU (which are clearly discussed) this is not possible. But it would be good to include mention of this as a possible error source and to calculate a ballpark estimate of this error. This can be done by looking at the IMU record from the pixhawk to identify the range of pitch/roll/yaw experienced during the flight and then calculating the impact of these values on the offset values. Also, a picture of the UAV setup showing this offset would help. Finally, I presume that the Y offset would flip depending which direction the UAV is flying? Is that the case? And if so how is it accounted for? Best practice for this design and without an IMU is to install the GNSS antenna directly above the camera so that only the Z offset is considered, and this is more or less static given vertical offsets are typically small.*

GPS/INS offset is implemented fully as a feature of Agisoft Photoscan and the reviewers' concerns are fully accounted for. X, Y, and Z here refer to a coordinate system with the point of origin as the camera with flight attitude taken into account (fig. 1). For more information see pages 56-57 of the Photoscan 1.4 User Manual (Agisoft, 2018).

[Figure]

**Figure 1** | *GPS/INS offset coordinate system. Figure taken from Photoscan documentation (Agisoft, 2018).*

We have adjusted the text in the methodology to clarify this coordinate system (P7L32).

*While this technique is relatively new in its application to glaciology, there have been a number of journal publications and grey literature that discuss the benefits and accuracy of GNSS-AT methods. The authors should include some of these citations and briefly mention this work in the introduction and/or discussion. Also some of these papers and industry grey literature compare GNSS-AT methods against more common GCP methods, which is arguably the best way to assess the accuracy of GNSS-AT. This is not done in this paper so would provide strong justification for the viability of the method.*

We agree that our paper is not the first to discuss the benefits of GNSS-AT as applied to UAV-based SfM, and did not intent to inadvertently imply that. The original contribution our paper makes with regards is as follows:

- The first application to glaciology, and one of the very few applications in geoscience generally.
- Implementation of GNSS-AT in a low-cost DIY frame, with a detailed and replicable description of methods.
- A modification of the method to allow for operation of a base unit on an advecting ice surface.
- Validation that the method is useful to assess ice dynamics via feature tracking methods.

On top of the papers we had already cited (Blankenberg, 1992; Hugenholtz *et al.* 2016; Benassi *et al.* 2017), we have incorporated further examples of GNSS-AT photogrammetry methods and accuracy assessment in the introduction (Mian *et al.* 2015, Fazeli *et al.* 2016). Applied use in the geosciences is currently extremely limited, but we have been able to find two examples of geosciences applications of GNSS-AT (in this case, RTK) outside of an accuracy assessment environment (Van der Sluijs *et al.* 2018, Strick *et al.* 2018).

We have also re-emphasised these papers in additional text in the introduction, introducing GNSS-AT firmly in the context of previous work, whilst highlighting the added value our work provides.

P3L1-13: Recent developments in lightweight, low-cost GNSS technology have allowed for the proliferation of a new technique whereby differential carrier-phase GNSS positioning is used to accurately geolocate imagery and subsequent photogrammetric products. This technique, known as GNSS-supported

aerial triangulation (GNSS-AT; Benassi et al., 2017), has been shown to result in sub-GSD horizontal errors without the use of GCPs (Mian et al., 2015; Fazeli et al., 2016; Hugenholtz et al., 2016; Benassi et al., 2017; van der Sluijs et al., 2018). Published applications of this technique in the geosciences are so far limited (van der Sluijs et al., 2018; Strick et al., 2018), and no studies yet examine the appropriateness of this technique for the study of glacial dynamics.

The aim of this paper is to: (i) apply GNSS-AT using a low-cost, custom-built airframe suitable for the study of extreme environments; (ii) develop and describe modifications to the GNSS-AT process to allow surveys to be undertaken at inland ice sheet location far from suitable GPS reference stations; and (iii) validate the method for the study of glacier dynamics.

In addition, we have included white papers by senseFly (Roze *et al.* 2014) and Wingtra (Ng and Buchheim, 2018) in later discussions of comparative GSD-scale uncertainties suggested by reviewer #1, although given their status as grey literature we have elected to not include them in the introduction.

*One final critique is the accuracy assessment. While I believe what was done is acceptable and does prove the method, it would be beneficial to have some sort of external comparison. Either by comparing the GNSS-AT method to the more typical (and accurate) GCP method, or to some other dataset, e.g. terrestrial/airborne LiDAR. This could be done over a small subset area where access is not as much of an issue – e.g. the bedrock zone.*

During our flights at the front, we did actually attempt to record validation GCPs. Here, we replicated the work of Ryan *et al.* (2015) and assessed a number of boulders across the northern side of the calving front visible in UAV orthophotos. Whilst this was adequate for error assessment when using the previous method of geolocation, our method was accurate enough that a the most we could say from this assessment was that our method was better than metre-scale (i.e. sub-boulder) accuracy (fig. 2). Whilst this was a pleasant surprise regarding the unexpected accuracy of the method, it did mean that we had to find alternative ways of estimating uncertainty, taking advantage of the number of repeat flights we made to assess variation rather than absolute error.

[Figure]

[Figure]

***Figure 2*** | *Example of validation GCP collection (left) and comparison against orthophoto (right) at calving front. Direction of photograph origin shown by white marker.*

We strongly agree that in an ideal world we would have access to a validation dataset. However, given the the remote location of the field site, and that our main site was on the ice sheet and not the front, there simply were limitations in what we could and could not achieve with the time available.

**Specific Comments**

*3:4 technically the second 'bedrock' exposure area is feasible to install GCPs, reword this accordingly.*

Reworded as follows:

P3L15 ...settings where on-ice GCPs are not feasible.

*3:5 prohibits should be prohibit*

Corrected

*3:12 at last 1948 should be at least 1948*

Corrected

*3:16 missing units, guessing km – should be "at least 30 km inland"*

Corrected

*4:7 Delete "a" – "the UAV is capable of ~1hour of flight time at ~60km"*

Corrected

*4:16 "allowing flight plans to avoid collision with cliffs" this sentence is awkwardly worded, perhaps rephrase*

Clarified as follows:

P4L32: ...avoiding collision with high relief topography at the glacier margins.

*4:18 "Out camera" should be "Our camera"*

Corrected.

*4:25 Great method! Is there a reason you didn't do this over the calving face as well?*

Thank you! Yes, there is. Our primary concern was one of flight endurance at the calving front (NB. the difference in scale between figs. 3 and 7). We chose to prioritise collecting four complete parallel flight paths of the entire width of the calving front over collecting oblique imagery. We considered that the far more extreme relief in the form of large crevasses and seracs at the front made additional oblique angles less essential than inland, where the surface was relatively flat. Briefly clarified in the text as follows:

P5L6: Flight paths in the ice sheet interior, where flight endurance allowed, also included a lower-altitude ~200 m along-track flightline...

*5:15 Repeats information presented briefly at 4:10 – Perhaps you can remove the text at 4:10 as an expanded discussion is included here.*

We have rewritten the two lines to emphasise hardware and components (former) and data production (latter) in an effort to reduce replication.

P4L21-23To allow for direct georeferencing of each photo location, we included an on-board lightweight L1 carrier-phase GNSS receiver (an Emlid Reach, using a small Tallysman TW4721 antenna with a 100mm ground plane).

P5L30-32: To obtain accurate camera positions we post-processed 5 Hz data logged by the on-board L1 carrier-phase GNSS receiver. Data were post-processed using...

*6:14 "was achievable" do you mean "was located nearby"*

Corrected

*6:14 If you used the EMLID reach in RTK mode you would be using the EMLID as the base station, correct? In which case the base station would only be single frequency. So this is another reason not to use the EMLID in RTK mode I believe. Or can EMLID interpret RTK corrections from other base station receivers (e.g. trimble etc?). On that note, I didn't see the make and model for the local dual frequency base station "B1". Also, reference station is trimble Net R9 receiver, but antenna not provided. Good to include these details if available.*

The Emlid Reach plays nicely with the open-source RTKLIB, so it should theoretically be possible (albeit probably finicky) to combine different hardware to achieve RTK results. But you would still need to know the base station position in real time to achieve absolute RTK positioning.

The base station (B1) and reference (B2) stations are as follows: we have clarified these in text.

- BASE - Trimble R9s receiver - Zephyr 3 Antenna
- REFERENCE - Trimble NetR9 receiver - Zephyr 3 Geodetic Antenna

P6L9-11: Clarified base and reference station models + antennas in text.

Thank you for highlighting this point - we have added it to the paragraph and rearranged it into a numbered list to make it comprehensible:

P7L3-6 ...we used instead post-processed kinematic (PPK) positioning for three primary reasons. … Second, PPK solutions are also often more accurate than RTK solutions as precise ephemeris data for the GNSS satellites is available during post processing.

Corrected.

Corrected.

Corrected.

*12:1 Couldn't you install the on ice base station for a few days (anywhere on the ice) and then process that data using the NRCAN PPP kinematic processing method (or OPUS) and then just pull the position of the base station at the time of the flight and use that as the static position for PPK processing of the UAV. That would mean you could work anywhere on the ice sheet and wouldn't be constrained to the 60km distance from the static reference station installed on bedrock.*

The reference to a ~60 km limit at this point in the text refers not to the reasonable baseline of dual-frequency processing but on the limit of reliable feature tracking given the observational errors in feature tracking versus daily glacier displacement.

Regardless, it is true to an extent that the base station could alternatively be positioned using kinematic PPP. However, whilst it is the case that PPP can achieve centimetre-level positioning given 24 hours of static observations (King *et al.* 2002), a fast-flowing glacier surface is not static over these timeframes (our on-ice base station was moving at a rate ~2 m $d^{-1}$). PPP kinematic processing for roving receivers is less accurate than kinematic relative differential carrier phase, achieving only sub-decimetre accuracy, and hence would have a downstream impact on photogrammetric product accuracy. Kinematic relative processing methods are modern standard best practice for rigorous in-situ GNSS data collection on the Greenland Ice Sheet, and with rigorous quality control have been used even up to 140 km from the ice margin/base station (Doyle *et al.* 2014). Further inland in an ice sheet environment, we would probably question the relevance of our technique anyway, as larger-scale seasonal patterns appear to dominate (*ibid.*). In an alternative slower-flowing environment, such as an alpine glacier, daily displacements are

probably low enough that a PPP kinematic method is a reasonable assumption. However, in such an environment, it would likely be the case that the base station is off the ice, in which case we would recommend falling back to the method we suggest for the calving front environment.

*14:4 the peak of which "is" centred*

Corrected.

*15:27 "moving one-ice GCP" should be "on-ice"*

Corrected.

*17:35 I think another point here is that dual frequency tends to provide more accurate positions on moving objects (i.e. UAV) especially in the vertical.*

Thanks for pointing out. This recommendation has been incorporated, alongside further changes to this paragraph recommended by reviewer #1, as follows:

L19P25-28 ...these GPS receivers are small and light enough to fit on small-sized UAV airframes, and hence allow for ... an improved flight baseline and accuracy

$$C_{pix} = \frac{s_{RMSE}}{\Delta x} - C_{match}$$

,

where $C_{pix}$ is estimate the horizontal uncertainty in pixels, $C_{match}$ is the uncertainty in the feature tracking algorithm, for which we use a typical value of 0.5 pixels, and $\Delta x$ is the raster resolution in metres. Hence, ($\sigma_{xy}$ can be calculated by multiplying $C_{pix}$ by the pixel resolution (0.2 m).) 
[revised manuscript text omitted]

---

## Author Response (AR2)

Tom Chudley
Scott Polar Research Institute
University of Cambridge
Lensfield Road
Cambridge CB2 1ER
United Kingdom

Tel: +44 (0) 1223 336574
Email: trc33@cam.ac.uk

5th March 2019

Dear Dr Karlsson,

**RE: Final Comments**

Thank you for your comments on our manuscript tc-2018-256 *High accuracy UAV photogrammetry of ice sheet dynamics with no ground control*.

We have responded to your comments below. Your comments are shown in italicised **blue**, our response in **black** and changes in manuscript in **red**. We have also attached a revised manuscript with highlighted changes: page numbers in the text below refer to this highlighted version of the manuscript.

We look forward to receiving a final decision on the manuscript shortly.

Sincerely,

Tom Chudley and Co-authors

**Response to Editorial Comments:**

*The use of m a.s.l. versus m above ground level: Presumably the latter is the parameter of interest but I have noted that both terms are used which is confusing to the reader (even if at least at the calving front they are almost identical). The term m a.s.l. is used in P1L11, P16L24 (in ref. to Ryan), P17L11 (twice in ref. to Jouvet), while m above ground level is found at P4L33 and P20L6. Please change for consistency.*

We have made this measurement consistent, using a.g.l. (above ground level) throughout the manuscript (P1L11, P4L26, P15L21, P16L8, P16L11, P16L29, P16L30, P19L17)

*P2L10: Is there a word missing here?*

We have slightly clarified the sentence - changed "is often impractical and a hindering factor needed to scale and orient" to "is often impractical, yet necessary to scale and orient" (P2L11-12)

*P2L17: The -1 should be redundant if the word "per" is included*

This had already been deleted (highlighted red in latexdiff).

*P4L21: This includes... (delete "is")*

Corrected (P4L15)

*P5L2: "...an auto.."*

Corrected (P4L28)

*P7L1: keep "correction"*

Corrected (P6L7)

*P13L7-12: I suggest moving this paragraph to the next section - or provide a better link to the preceding paragraphs. As it reads now the mention of measurements in the ice sheet interior is confusing and seemingly out of the blue.*

We have moved second half of this paragraph, with minor textual changes, to the beginning of the next section (3.2) (P11L34-35; P12L1-11)

*P13L28: separation*

Corrected (P15L2)

*P16L9: Missing a word? "down to"?*

Corrected (P15L27)

*P16L23: receiver*

Corrected (P16L7)

*P17L10: missing "as": "as well as"*

Corrected (P16L28)

[revised manuscript text omitted]